# New estimates of pan-Arctic sea ice–atmosphere neutral drag coefficients from ICESat-2 elevation data

Alexander Mchedlishvili[1], Christof Lüpkes[2], Alek Petty[3,4], Michel Tsamados[5], and Gunnar Spreen[1]

[1]Institute of Environmental Physics, University of Bremen, Bremen, Germany
[2]Alfred Wegener Institute for Polar and Marine Research, Bremerhaven, Germany
[3]Goddard Space Flight Center, National Aeronautics and Space Administration, Greenbelt, MD, USA
[4]University of Maryland, College Park, MD, USA
[5]Department of Earth Sciences, University College London, London, UK

**Correspondence:** Alexander Mchedlishvili (alexander.mchedlishvili@uni-bremen.de)

**Abstract.** The effect that sea ice topography has on the momentum transfer between ice and atmosphere is not fully quantified due to the vast extent of the Arctic and limitations of current measurement techniques. Here we present a method to estimate pan-Arctic momentum transfer via a parameterization which links sea ice–atmosphere form drag coefficients with surface feature height and spacing. We measure these sea ice surface feature parameters using the Ice, Cloud and land Elevation Satellite-2 (ICESat-2). Though ICESat-2 is unable to resolve as well as airborne surveys, it has a higher along-track spatial resolution than other contemporary altimeter satellites. As some narrow obstacles are effectively smoothed out by the ICESat-2 ATL07 spatial resolution, we use near-coincident high-resolution Airborne Topographic Mapper (ATM) elevation data from NASA's Operation IceBridge (OIB) mission to scale up the regional ICESat-2 drag estimates. By also incorporating drag due to open water, floe edges and sea ice skin drag, we produced a time series of average total pan-Arctic neutral atmospheric drag coefficient estimates from November 2018 to May 2022. Here we have observed its temporal evolution to be unique and not directly tied to sea ice extent. By also mapping 3-month aggregates for the years 2019, 2020 and 2021 for better regional analysis, we found the thick multiyear ice area directly north of the Canadian Archipelago and Greenland to be consistently above $2.0 \cdot 10^{-3}$ while most of the multiyear ice portion of the Arctic is typically around $\sim 1.5 \cdot 10^{-3}$.

## 1 Introduction

Arctic sea ice is heterogeneous with respect to several characteristics including its concentration, thickness, and roughness (e.g., Thorndike et al., 1975; Bourke and Garrett, 1987). The understanding of how these parameters vary with space and time is important for several reasons, including its impact on human activities, e.g. navigation, and its impact on the physical system, e.g. the transfer of momentum and energy between the atmosphere and ocean.

Studies of Arctic sea ice have arguably focused more on constraining variability in concentration and thickness, towards estimating sea ice volume and its variability in time and space. However, the surface roughness of sea ice also exhibits strong spatial and temporal variability (e.g., Andreas et al., 2010; Lüpkes et al., 2012; Castellani et al., 2014; Petty et al., 2017) which needs to be better understood. Surface roughness can be related to the neutral drag coefficient by applying the Monin-Obukhov

theory. Since the roughness length for momentum and the scalar roughness length for heat and moisture occur also in the non-neutral transfer coefficients, surface roughness directly impacts not only momentum transport but also the transfer of heat and moisture between the atmosphere and the underlying surface. Rougher surfaces can create more turbulence and enhance mixing, thereby influencing the stability of the atmospheric boundary layer (e.g., Garratt, 1992; Schneider et al., 2022; Lüpkes and Gryanik, 2015). Due to its impact on momentum and heat transport over and below the sea ice layer, surface roughness is a fundamental parameter influencing the distribution of sea ice (e.g., Yu et al., 2020; Brenner et al., 2021). Both its relevance for heat and moisture transfer as well as momentum transfer are described by the Monin-Obukhov theory for the determination of turbulent fluxes, where surface roughness serves as an essential parameter (e.g., Lüpkes et al., 2012; Lüpkes and Gryanik, 2015). The process of becoming rough is driven in-part by pressure ridging, which redistributes ice vertically, as well as the presence of snow features such as dunes and sastrugis (e.g., Arya, 1975; Hopkins, 1998; Petty et al., 2016). Summer melt can facilitate the smoothing of obstacles like ridges but can also increase roughness through ice melt (e.g., Andreas et al., 2010; Landy et al., 2015; Castellani et al., 2014). Rougher sea ice is generally found in the thick, multiyear ice regions north of the Arctic Archipelago and Greenland. Landfast rough ice in these areas is an important factor for determining transportation routes for local residents and industry (Dammann et al., 2018). Newly formed first-year ice is typically much smoother, although this division is complicated by the accumulation of snow and its ability to smooth out ice surface variability (e.g., Garbrecht et al., 2002). Observational (e.g., Castellani et al., 2014) and model studies (e.g., Tsamados et al., 2014) suggest that sea ice roughness varies more with space, e.g. between first-year ice and multiyear ice regions, than it does with time, e.g. between freeze-up and melt seasons. With the decline of rough multiyear ice due to recent sea ice minima (e.g., Stroeve et al., 2012), the Central Arctic as well as areas north of Eurasia and Alaska are predominantly composed of first-year ice during winter months and are therefore smoother in comparison (e.g., Castellani et al., 2014; Petty et al., 2017).

The roughness of sea ice is heavily linked with its motion as a result of momentum and energy transfer between ocean, sea ice and atmosphere. Disregarding the proportionally little Arctic sea ice that is fastened to the surrounding landmasses, the remaining majority is subject to motion from the balance of drag forces from ocean currents and winds as well as internal forces (e.g., Thorndike and Colony, 1982; Steele et al., 1997). Ice motion redistributes ice and snow around the Arctic and controls the rate of discharge from the Arctic basin. The balance of forces governing this motion is described in the momentum balance equation for sea ice, in which the interactions between ice, atmosphere and ocean are quantified via drag coefficients. The turbulent surface flux of momentum $\boldsymbol{\tau}$ that describes this interaction is as follows

$$\boldsymbol{\tau} = \rho C_d(z) U(z) \left( \cos\theta\, \boldsymbol{U}(z) + \sin\theta\, \hat{k} \times \boldsymbol{U}(z) \right) \tag{1}$$

where $\rho$ is the air density, $\boldsymbol{U}(z)$ is the difference in air and ice velocities at a given height $z$ and $U(z)$ is its magnitude, $\hat{k}$ is the vertical unit vector and $\theta$ is the turning angle. The drag coefficient $C_d$ is usually written as a product of the neutral drag coefficient $C_d^n$ and a surface roughness dependent stability function $f_m$ (e.g., Garratt, 1992; Birnbaum and Lüpkes, 2002; Lüpkes and Gryanik, 2015; Gryanik and Lüpkes, 2018). The height above sea level $z$ is most commonly set to a reference height of 10 m to match the layer for which the Monin-Obukhov theory for the determination of turbulent fluxes is valid. It is furthermore nearest to the lowest height level of high-resolution climate and weather prediction models. The neutral drag

coefficient $C_d^n$ assumes a neutrally stratified atmospheric surface layer, and is the key parameter that is investigated in this study. The total drag coefficient over a given sea ice surface is commonly subdivided into a contribution from skin drag $C_{d,s}$, caused by microscale roughness, and a contribution by form drag $C_{d,f}$, caused by large distinct obstacles (Arya, 1973, 1975).

This division is the basis for the drag parameterization developed by Garbrecht et al. (2002). The derived parameterization is developed for estimating the form drag component of the total neutral 10 meter drag coefficient $C_{d10,f}^n$ from the distribution of distinct obstacles and their heights relative to the surface.

The difference in air and ice velocities $\boldsymbol{U}(z)$ varies in space and time and thus also the associated momentum transfer and drag forces show a corresponding variability. Given the changing Arctic climate, and the above-mentioned shift from multiyear
ice to first-year ice, we can expect that, with a changing distribution of spatial roughness, the associated drag forces will change also. It is therefore in our best interest to help constrain drag coefficients to better model sea ice–atmosphere momentum transfer and in turn, the Arctic climate system. In this study, we will be focusing on the interaction between the atmosphere and sea ice and the related atmospheric (wind) drag force, but will avoid extrapolating our findings to the equally important interaction between ocean and sea ice since our observations are limited to satellite and airborne measurements.

The Garbrecht et al. (2002) parameterization, discussed further in subsequent sections, has been used in various Arctic regions using airborne topographic data (Castellani et al., 2014; Petty et al., 2017). We now aim to extend the applicability of said parameterization onto the high-resolution pan-Arctic topographic data measured by the Advanced Topographic Laser Altimeter System (ATLAS) onboard NASA's Ice, Cloud and land Elevation Satellite-2 (ICESat-2) and better characterize the spatio-temporal variability in form drag. With this data product we hope to aid the development of future climate models with
integrated form drag parameterization schemes (e.g., Tremblay and Mysak, 1977; Steiner et al., 1999; Tsamados et al., 2014; Yu et al., 2020; Elvidge et al., 2021). Model studies with integrated form drag schemes have been shown to better characterize both ice–atmosphere and ice-ocean interactions as well as inherent properties of sea ice like its thickness (e.g., Tsamados et al., 2016; Martin et al., 2016). However, the degree of uncertainty remains large primarily due to a lack of constraints in these form drag parameterization schemes. While airborne topographic data is perhaps the best record of measured sea ice drag
coefficients in the Arctic, it cannot reliably be used to constrain model drag coefficients because of its incomplete temporal and spatial coverage. Satellite drag coefficient evaluations using topography data, on the other hand, have in the past been impractical due to their inability to detect sea ice roughness on sufficiently small scales (e.g., Landy et al., 2015; Castellani et al., 2014). With the launch of NASA's ICESat-2 laser altimeter satellite in 2018, which is able to collect topographic data over sea ice at a resolution that is higher than its predecessors (10s of meters - able to resolve distinct sea ice features), this
study aims to make use of the advancements in satellite altimetry to estimate the neutral drag coefficients across the entire Arctic and highlight its regional and seasonal variability for the first time.

## 2   Data and Methods

This section describes the data-sets involved in this study as well as the parameterizations used to calculate drag coefficients.

## 2.1 ATLAS on ICESat-2

The Advanced Topographic Laser Altimeter System (ATLAS) is a lidar instrument onboard ICESat-2 that collects high resolution surface elevation data using a sophisticated split-beam photon counting laser system (Neumann et al., 2019). By determining the travel time of reflected laser pulses, ATLAS is able to accurately measure small changes in topography through differences in along-track elevation. The six laser beams are divided into three beam pairs consisting of a strong and a weak beam. The separation between each of the beam pairs is about 3.3 km across-track, whereas the separation between the strong and weak beams making up the pairs is 2.3 km along-track and 90 m across-track (Markus et al., 2017). At an altitude of 500 km, the 10 kHz laser pulses that ATLAS transmits result in roughly 11 m diameter laser footprints (Magruder et al., 2020, 2021) that are spaced 0.7 m apart. Here what we refer to as a footprint is the spatial extent of the laser energy illumination on the observed surface (Magruder et al., 2020).

In this study we focus on the along-track heights for sea ice and open water leads (ICESat-2 ATL07/L3A level 3a data product). ATL07/L3A takes the global geolocated photon data (ATL03/L2) as input and further processes it to obtain information on sea ice topography (e.g., Kwok et al., 2021a, 2019b). For each of the six laser beams, estimates of sea ice surface heights are computed by applying various filters (to remove background photons) and a dual-Gaussian fit to segments of varying length, over which 150 signal photons are accumulated. This is done to reduce the vertical errors from $\sim$30 cm for each photon height to $\sim$2 cm (over flat surfaces) for each ATL07 segment height (Kwok et al., 2019a). The segment length varies based on surface type which influences photon counts such that when photon counts are low, segment length is high and vice versa (Kwok et al., 2021a). The spatial resolution is then the sum of the segment length and beam footprint and are on average $\sim$ 30 m for the strong beams and $\sim$ 70 m for the weak beams (Kwok et al., 2019a). The strong beams (beams 1, 3 and 5) are roughly 4 times stronger in terms of pulse energy than the weak beams (beams 2, 4 and 6), which results in these segment length differences (Kwok et al., 2019b). As a result, we only use the three strong beams for this study to take advantage of the better resolution that they provide. The retrieved heights are referenced to the WGS84 ellipsoid and include various geophysical corrections (e.g. ocean tides, inverted barometer, mean sea surface). ATLAS distinguishes water and ice surfaces by utilizing the surface photon rate, the width of the photon distribution and background rate (Kwok et al., 2021a). ATL07 is also restricted to regions of ice concentration greater than 15% based on passive microwave data.

The ICESat-2 level-2 geolocated photon product ATL03 provides data at higher spatial resolution than the aggregated ATL07 dataset, but at the cost of reduced precision and vastly increased data volume. The use of ATL03 and higher-resolution along-track aggregations has been shown to help better detect and resolve distinct pressure ridge sail heights compared to ATL07 (Ricker et al., 2023). However, for this study, we opted to use the more readily available ATL07 dataset together with our Operation IceBridge downscaling to ensure that our ridge detection and form drag methodology can directly be applied to all existing and upcoming ICESat-2 sea ice elevation data.

## 2.2  ATM Lidar on Operation IceBridge airplanes

The Airborne Topograhic Mapper (ATM) is an instrument suite that contains two high-resolution conically scanning laser altimeters at 1.5° and 2.5° off-nadir, able to measure surface elevation with swath widths 245 m and 40 m, respectively (Studinger, M. 2013, updated 2020; MacGregor et al., 2021; Studinger et al., 2022). Like ATLAS, the lidar uses a 532 nm laser (narrow scanner also uses 1064 nm laser) and a 10 kHz pulse repetition frequency with each laser spot having a footprint of ~1 m and a vertical precision of ~10 cm (Martin et al., 2012; Studinger et al., 2022). Here we use the wide scanner (1.5° off-nadir) to take advantage of its high data density at the edges of the swath. NASA carried out several airborne campaigns in the Arctic and Antarctic named "Operation IceBridge" (OIB) during recent years targeting land and sea ice observations (MacGregor et al., 2021). Throughout OIB the ATM lidar instrument has been installed aboard and carried by NASA aircrafts (NASA P3-B and NASA DC8 aircrafts) (Kwok et al., 2019a).

The OIB ATM data-set used in this study is from April 2019 when 4 of the flights over sea ice are near-coincident in space and time with ICESat-2. This includes all data from 8, 12, 19 and 22 April 2019 throughout which coincidence was variable but sufficient for comparing observations of similar ice regimes (Kwok et al., 2019a). This data-set is used to derive a scaling factor via regression with ICESat-2 ATL07-derived drag coefficients as it is has a better spatial resolution and therefore better resolves sea ice features. By applying this factor to the ICESat-2 ATL07-derived drag coefficients using near-coincident OIB ATM data, we hope to mitigate the spatial sampling biases discussed in section 3. In addition, the 6 and 20 April 2019 flights across sea ice that were not coincident with any ICESat-2 tracks, are used to independantly evaluate our ICESat-2 ATL07 monthly pan-Arctic neutral drag coefficient estimates. The OIB flight lines are outlined in Fig. 3A of section 3.

## 2.3  Extracting sea ice feature data

The Garbrecht et al. (2002) sea ice drag parameterization requires obstacle (sea ice feature e.g., a pressure ridge, rubble field, hummock, snow dune, sastrugi) height and spacing for the calculation of drag coefficients. Regional averages of these quantities are derived from the ATL07 data over segments that are chunked prior to the sea ice feature extraction (please see Fig. A8 in the appendix, where this chunking procedure, as well as the processing steps that follow, are depicted and further described). After experimenting with multiple segment sizes over which to calculate those regional averages, 10 km segments were chosen as in Castellani et al. (2014). 10 kilometres is a typical grid length of a high-resolution regional climate model and is proposed to be a reasonable minimum for the drag parameterizations used (Garbrecht et al., 2002; Lüpkes et al., 2012). Importantly, the data is not equally spaced due to the influence of surface reflectivity on photon counts and thus the along-track distance parameter (in meters) is used to chunk the data into 10 km segments. As a result, the 10 km segments end up having a similar but not equal amount of values. To see the typical spacing between measurements Arctic-wide and the spatial variability thereof, see Fig. A1B in the appendix. Lastly, to increase the number of segments and the stability of drag estimates, the 10 km segments over which average sea ice feature statistics are calculated are shifted by 1 km along-track, i.e., they have large overlap and only every 10th segment is fully uncorrelated. Importantly, 10 km segments with a measurement spacing that exceeds 1 km, a value

that is sufficiently higher than what can be attributed to dark non-reflective surfaces, are assumed to have cloud contamination and are therefore omitted.

After segmentation, the surface level is subtracted from all values per 10 km segment. While the surface height estimates are referenced with respect to the mean sea surface (Kwok et al., 2021a), the drag calculations require obstacle heights above level ground and not the sea surface (freeboard). Thus the sea ice heights are first binned (rounded to the nearest centimeter) and then the height of the surface level is calculated by computing the mode for all heights within the 10 km segment. By subtracting this mode, all heights are referenced to the regional ice level surface. For bimodal distributions, the higher mode is used so as to avoid modes associated to leads and young ice.

The produced 10 km segments of elevation from the regional sea ice surface are used to calculate average obstacle height and obstacle spacing per segment. A four-step process is applied to each segment to compute these regional parameters.

1. The first step involves finding local maxima, i.e. obstacle heights along the segment.

2. The second step is to omit obstacle heights that are below a chosen threshold value.

3. The third step is to distinguish individual features by omitting obstacles that do not fulfil the Rayleigh Criterion (explained below).

4. Finally the fourth step is to compute the spacing between the obstacles that fulfil the Rayleigh Criterion.

The Rayleigh Criterion states that two maxima (obstacles) must be separated by a minimum that is less than half the value of the higher maxima for them to be classified as two separate features (e.g., Hibler, 1975; Wadhams and Davy, 1986). After omitting all elevation maxima that do not fulfil the Rayleigh Criterion, the obstacle heights and the spacing between them (both in meters) are averaged over each 10 km segment, before calculating the neutral drag coefficients at this same scale.

While the chosen threshold value of 0.2 m elevation is expected to detect not only pressure ridges but also all topographic features like rubble fields and hummocks, here we define an obstacle as any series of connected elevation values above the cutoff. This is because all obstacles have the ability to impart form drag and it is therefore not necessary to distinguish between them. Nevertheless, some cutoff must be introduced to effectively partition centimeter-scale roughness that is associated to skin drag and form drag associated to obstacles (in this case anything above the 20 cm cutoff), and we chose one which has been used before (e.g., Castellani et al., 2014; Petty et al., 2017) for better a comparison with previous evaluations of Arctic sea ice topography. A more pressure ridge-focused threshold value of 0.8 m (used alongside 0.2 m in Castellani et al. (2014)) was also tested and produced similar results (not shown).

As will be shown in section 3, the higher resolution OIB ATM data, that is able to better resolve sea ice features, is used to bias correct and account for sampling differences in ICESat-2 ATL07 data. Prior to extracting sea ice features, the conically scanned along-track topographic two dimensional data from ATM must also be converted into a one dimensional track. To do so, we adopt the methods from Petty et al. (2017) wherein using a given azimuth angle range we can isolate different parts of the conically scanned ATM swath. We use the ranges 355 to 5 ° and 175 to 185 ° to extract the outermost narrow parts of the full ATM swath with the highest data density. This narrow track is then ordered as a function of distance from the first data

point and interpolated to fix the resolution at 1 m along-track. Once ordered, as with ICESat-2 ATL07 elevation data, the OIB ATM data set undergoes the 10-km chunking and the four-step process outlined above.

The OIB ATM high-resolution airborne data-set is then processed and drag coefficients are calculated from the sea ice feature statistics obtained for 10 km segments (see section 2.4 for more about the calculation step). The processed OIB ATM data serves as an independent drag coefficient estimate to which we compare the processed ICESat-2 ATL07 data. The comparison is done on a regional scale by binning both output datasets onto a polar stereographic projection grid with nominal gridded resolution of 12.5 km. The Polar Stereographic projections is true at 70 degrees north with up to 6% distortion at the poles (Knowles, 1993); making it an ideal candidate for pan-Arctic maps. The resampling step for the two datasets is done to compare drag coefficients averaged over the same area; this is because the 10-km segments are not perfectly aligned with one another. Once all coincident grid-cells are identified the bivariate distribution of the two gridded data products is generated.

We use the OIB ATM data as reference to account for the spatial sampling differences with the lower resolution ICESat-2 data. A Huber Regressor is calculated from filled grid cells of near-coincident data and model parameters are then used to linearly scale up the ICESat-2 ATL07 drag coefficient estimates. Unlike the traditional linear fit, the Huber Regressor applies a linear loss to samples with an absolute error $|z|$ larger than a given threshold value $\epsilon$ (set to $1.35$ to achieve 95% statistical efficiency), thereby weighting 'inliers' and 'outliers' differently (Huber and Ronchetti, 2009). This is done to reduce the sensitivity of the loss function to outliers that are expected in the data due to the high level of uncertainty when comparing quantities averaged over large spatial scales. Importantly, OIB ATM data is taken as the independent true variable upon training the model as ICESat-2 ATL07 is expected to underestimate obstacles due its lower spatial resolution and therefore overestimate obstacle spacing because of the cutoff.

## 2.4 Calculating neutral form drag coefficient

With the extracted sea ice feature statistics, we apply to them the form drag parameterization developed in Garbrecht et al. (2002). The parameterization is based on the formulation of Garbrecht et al. (1999), which itself is built upon findings by Arya (1973, 1975) and Hanssen-Bauer and Gjessing (1988) on momentum fluxes by single obstacles. While there are other parameterizations of surface drag (e.g., Lüpkes et al., 2012, 2013; Tsamados et al., 2014), here we focus on the one by Garbrecht et al. (2002) as it is optimized for one-dimensional data like ICESat-2 ATL07 and better suited for estimating drag due to obstacles over consolidated ice-cover.

The generalized Garbrecht et al. (2002) formulation for the atmosphere-ice form drag coefficient is as follows

$$C_{d,z_r,f} = \frac{1}{2}\frac{c_w}{\Delta x}\left[\frac{1}{\ln(z_r/z_0) - \Psi(z_r/L)}\right]^2 \int\limits_{z_0}^{H_r} [\ln(z/z_0) - \Psi(z/L)]^2 \, dz \qquad (2)$$

where $c_w$ is the coefficient of resistance, $z_r$ is the reference height, $z_0$ is surface roughness length and $\Psi(z/L)$ is the Monin Obukhov stability correction function. $H_r$ and $\Delta x$ represent obstacle height and obstacle spacing respectively which, as in Garbrecht et al. (2002), we will generalize to ensemble mean values $H_e$ and $x_e$. We use a 10 m reference height $z_r$ since it is the widely accepted value and is often the lowest level available from atmospheric models. Computing drag coefficients without

knowing the orientation of obstacles brings with it its own uncertainty and the Garbrecht et al. (2002) parameterization accounts for this problem by reducing the form drag by a factor of $2/\pi$ given the assumption that obstacles are oriented randomly (Mock et al., 1972). An uncertainty of roughly $\pm20\%$ is introduced on account of this assumption (Castellani et al., 2014). Lastly, to simplify further, we estimate the atmospheric neutral drag coefficient $C_d^n$ only and do not consider the stability correction. The effect of the latter on form drag is explained in Birnbaum and Lüpkes (2002) and in more detail by Lüpkes and Gryanik (2015). With all the caveats taken into account and the integral having been evaluated we get the equation as presented in Castellani et al. (2014):

$$C_{d,10,f}^n = \frac{c_w H_e}{\pi x_e} \frac{[\ln(H_e/z_0) - 1]^2 + 1 - 2(z_0/H_e)}{[\ln(10/z_0)]^2} \tag{3}$$

This equation goes back to Garbrecht et al. (2002), but since only neutral atmospheric stability conditions are considered, the integrals in the corresponding equation can be solved analytically. Averaged obstacle height $H_e$ and spacing $x_e$ are the two parameters that are extracted from the ICESat-2 sea ice height data as mentioned in the section before. Here we use the Garbrecht et al. (2002) formulation for the coefficient of resistance and compute it as a function of obstacle height $c_w = 0.185 + 0.147 H_e$, where $0.147$ is in $\mathrm{m}^{-1}$ so that $c_w$ is unitless.

To calculate the total neutral drag coefficient $C_{d,10,s}$ we follow (Arya, 1973, 1975) and add the skin drag coefficient using a value that has been derived by Garbrecht et al. (2002) from airborne turbulence measurements over very smooth sea ice. They obtained the value $8.38 \times 10^{-4}$ by use of

$$C_{d,10,s}^n = \left[ \frac{\kappa}{\ln(10/z_0)} \right]^2 \tag{4}$$

with the von Kármán constant $\kappa = 0.4$ and $z_0 = 1 \times 10^{-5}$ m. This value has its own associated uncertainty (see Section 3.6). It is important to note that equation 2, and by extension equation 3, are only valid with the assumption of obstacle spacing being large enough such that the flow can return to its undisturbed state in between obstacles (Garbrecht et al., 2002). In Garbrecht et al. (2002), although the critical value of $H_e/x_e = 0.015$ for which this condition is satisfied was exceeded by the observed aspect ratio, the parameterization that accounts for this effect (Hanssen-Bauer and Gjessing, 1988) was neglected since the resulting form drag $C_{d,10,f}$ was changed by less than 3%. Similarly, in Castellani et al. (2014), including the sheltering effect leads to a modification of less than 0.05% of the total drag coefficient. In this study, the sheltering function $(1 - \exp(-0.5x_e/H_e))^2$ (e.g., Hanssen-Bauer and Gjessing, 1988; Lüpkes et al., 2012; Castellani et al., 2014) is multiplied by OIB ATM-derived form drag coefficient estimates (derived via equation 3) but not ICESat-2 ATL07 data since due to the smoothing effect overestimating obstacle spacing (discussed further in section 3), the aspect ratio ends up being predominantly less than 0.015 Arctic-wide for ICESat-2. Despite this, we did conduct our own sensitivity study to see the effect of the sheltering function on ICESat-2 ATL07 topography data and elaborate further on this topic in section 3.6.

## 2.5 Calculating total neutral drag coefficient

As the last step in our study, we also included the skin drag coefficient of open water $C_{d,10,ow}^n$ and the form drag coefficient of floe edges $C_{d,10,e}^n$. For the drag coefficient over open water $C_{d,10,ow}^n$, we use the constant value $1.5 \cdot 10^{-3}$ which is multiplied by $(1-A)$ where $A$ is the sea ice concentration. $C_{d,10,e}^n$ is implemented using the parameterization of form drag by floe edges of Lüpkes et al. (2012), given in the most simplified form (hierarchy level 4) as $C_{d,10,e}^n = 3.67(1-A) \cdot 10^{-3}$ where the latter term $(1-A)$ when multiplied by $A$, peaks at 50% sea ice concentration signifying areas with both ice and water. The parameterization does not just represent a simple fit to observations but was rather derived from physical concepts and assumptions based upon the drag partitioning scheme by Arya (1973, 1975); for further information please see Lüpkes et al. (2012). We use sea ice concentration from the AMSR2 microwave radiometer at $6.25\,\text{km}$ grid resolution based on the ASI algorithm (Melsheimer and Spreen, 2019; Spreen et al., 2008). The combined equation for the neutral 10 m sea ice–atmosphere drag, taken from Petty et al. (2017), is then as follows

$$C_{d,10,T}^n = (1-A)C_{d,10,ow}^n + A\left(C_{d,10,s}^n + C_{d,10,e}^n + C_{d,10,o}^n\right) \tag{5}$$

where $C_{d,10,o}^n$ is the form drag coefficient caused by obstacles (e.g. pressure ridges, sastrugis) calculated from ICESat-2 elevation data with equation 3. All terms of equation 5 are referenced to a height of 10-m and thus so is $C_{d,T}^n$. Equation 5 is evaluated on daily ICESat-2 ATL07 tracks and we match daily ASI sea ice concentration maps to the ICESat-2 ATL07 tracks for the given day to ensure consistent sampling approaches from the different data-sets.

## 3 Results and discussion

The four-step process explained in section 2.3 is evaluated on near-coincident OIB ATM data (A,B) and ICESat-2 ATL07 data (C) in Fig. 1. Local maxima are found and those below the threshold of 20 cm (marked with a red dashed line in the figure) are omitted (maxima that above the threshold are marked with a filled-in black circle in the figure). Thereafter, the Rayleigh criterion is evaluated (those that fulfil the criterion are marked with a yellow 'x'). That is why we see a lot of unmarked black circles on the side of obstacles, because the Rayleigh criterion assures that only the maximum of the whole feature is considered (most clearly visible in Fig. 1A). Figs. 1A and 1B both depict the same 1 km long ATM segment from an OIB flight carried out on the 19th of April, 2019. The segment chosen is along the 88th parallel north and spans the longitude range 170.60-170.85°E, putting it firmly within the central Arctic. The difference between Figs. 1A and 1B is that 1B has a moving average filter of box size 30 m applied. This is done to simulate the 30 m ICESat-2 ATL07 footprint (see section 2.1) which, as a result of the dual-Gaussian fit needed to reduce vertical uncertainty, also in effect smooths out the topography. For a more detailed description and case study of this smoothing effect, the reader is referred to the publication by Ricker et al. (2023). Once the topography data is smoothed using this 30 m box filter, small clusters of narrow obstacles are viewed as one and the average distance between them for a given length scale is enlarged. In the case presented, average obstacle spacing $x_e$ increases by a factor of $\sim 5.2$. Average obstacle height $H_e$ comes out at 0.35 m for both plots. While the maximum obstacle height is larger in the original data, the smoothed data also has a smaller number of shorter obstacles that bring down the average. In general,

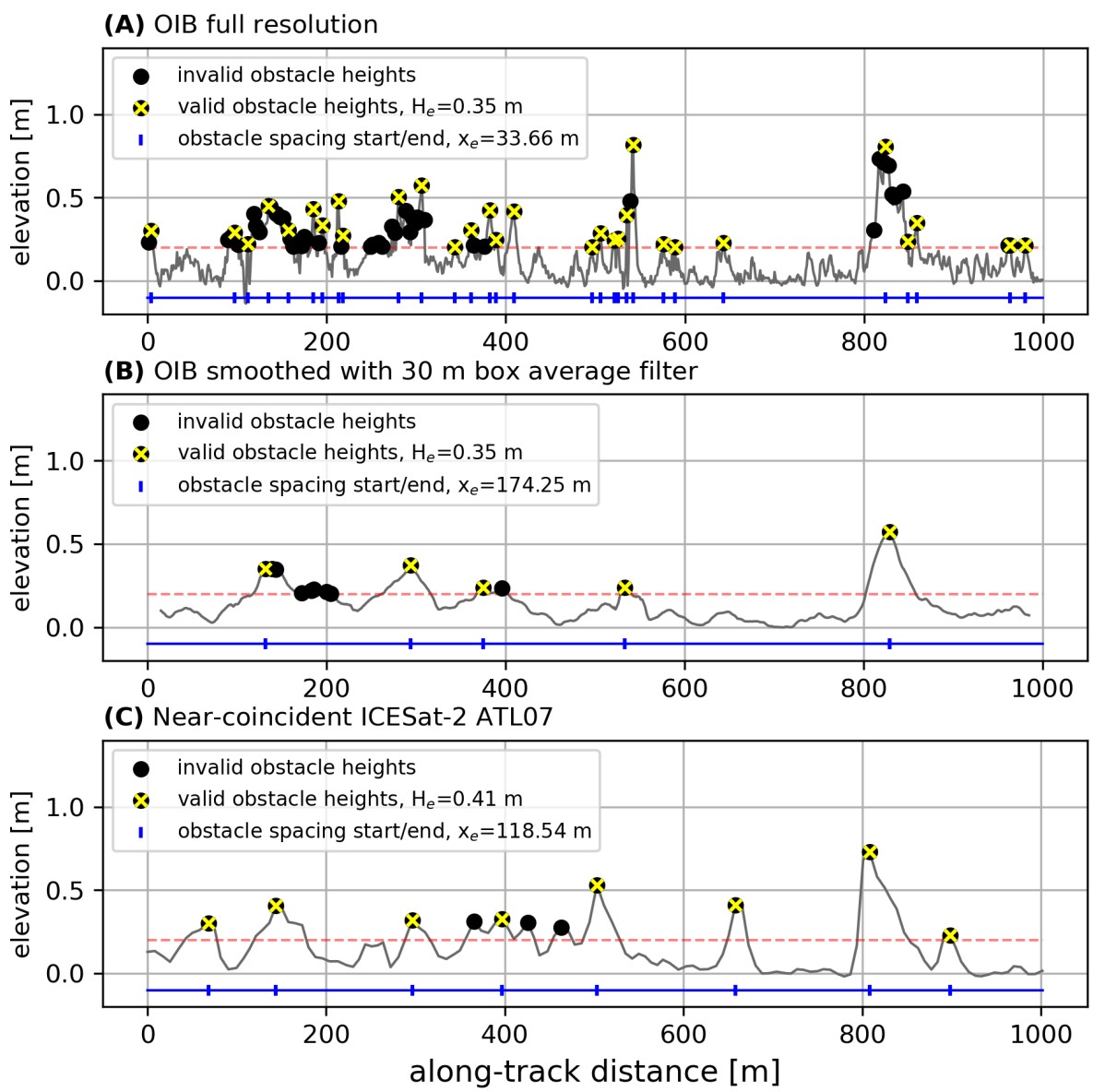

**Figure 1.** (A) Sea ice feature statistics from a sample OIB ATM flight on 19 April 2019 ($\sim$170.7°E, 88.0°N). (B) Same as (A) but smoothed to the ICESat-2 resolution via a moving average filter with box size of 30 m. (C) Sea ice feature statistics from a near-coincident ICESat-2 ATL07 track section. Black dots: all identified maxima; yellow 'x': maxima which satisfy the Rayleigh Criterion; red dashed line: 0.2 m threshold; blue line with dividers: identified obstacle spacing. All data is referenced to level ice (mode).

we expect the height of tall narrow ridges to be underestimated due to sampling however. We can observe the smoothing effect in Fig. 1C wherein near-coincident ICESat-2 ATL07 data, with its low spatial resolution relative to OIB ATM, also exhibits
larger average obstacle spacing (factor of ∼3.5) and therefore lower drag coefficient. As Fig. 1 covers only a small distance of 1 km to demonstrate the obstacle peak finding method, values presented are likely not representative of all data.

### 3.1   Drag coefficient regression with airborne lidar measurements

Taking the spring 2019 OIB/ICESat-2 underflights (4 days in 2019) that were near-coincident with the measurements of ICESat-2, we can calculate drag coefficients from both data sets and compare the results. The shortest time-lag during the
underflight was less than 1 min for 3 of the flights (8, 12 and 19 April 2019) (Kwok et al., 2019a), however, 8 and 12 of April are overlapping racetracks conducted for a time period of ∼8 hours, thus the time-lag is highly variable (all legs were considered to maximize the total amount of data). The shortest time lag on April 22, also used for this comparison, was ∼38 min (Kwok et al., 2019a). We used a subset of all OIB data that fell within the specified azimuthal angle range of the ATM scanner which likely reduced the spatial coincidence as well. As a result, we did not simulate a one to one elevation comparison as
has already been done in Kwok et al. (2019a) and Ricker et al. (2023) and thus did not employ any drift correction. Since we look at 10-km averages for the purpose of comparing regional average form drag, it was sufficient to compare averaged data of similar ice regimes and not to focus on the coincidence itself. Since the averages are not aligned, the data-sets are gridded to a 12.5 km grid and the comparison takes place between matching filled-in grid cells (see Fig. 2).

Thus Fig. 2 shows a comparison between form drag coefficients calculated from ICESat-2 ATL07 and OIB ATM segments
(blue). This slope (in blue) is the scaling that is applied to the ICESat-2 ATL07 drag coefficients to amplify the retrieved signal, while the orange, green and pink lines are simply tests done to better explain the relation between the satellite and airborne data-sets. As expected, the majority of form drag coefficients calculated from OIB ATM occupy a wider range ($\sim 0.3 - 1.3 \cdot 10^{-3}$) than their ICESat-2 ATL07 grid cell counterparts ($\sim 0 - 0.3 \cdot 10^{-3}$). As demonstrated in Fig. 1, we can simulate ICESat-2 ATL07 by passing all OIB ATM data through a moving average filter of varying box sizes (15 m, 30 m and 45 m) and observe
that we can get the line of best fit to match the one-to-one line depending on the size of the averaging box (Fig. 2). Smoothing with a box size of 30 m, which is comparable to the ATL07 strong beam $\sim 30$ m spatial resolution (Kwok et al., 2019b), results in a line of best fit that is the closest match to the one to one line, which is encouraging. Box sizes 15 m and 45 m are shown for comparison's sake and are meant to demonstrate how both too little and too much of this smoothing can fail to produce values comparable to that of ICESat-2.

The beam used for model training is the second strong beam as it is in best spatial agreement with all 4 OIB ATM flight near-coincident data (Kwok et al., 2019a). Using the line of best fit from Fig. 2 (in blue) we correct the ICESat-2 ATL07 form drag coefficients towards the OIB ATM form drag coefficient range using the scaling factor 5.28. Here we focused on comparing the average drag coefficients from satellite and airborne instruments rather than the component parameters: obstacle height and obstacle spacing. The reason for this approach is because that is where the best regression was found. Regressing obstacle
heights shows decent agreement but evaluating the different box sizes on the OIB values shows very small differences. The differences are small because while the smoothing introduced in ATL07 effectively retrieves the tall narrow ridges as smaller

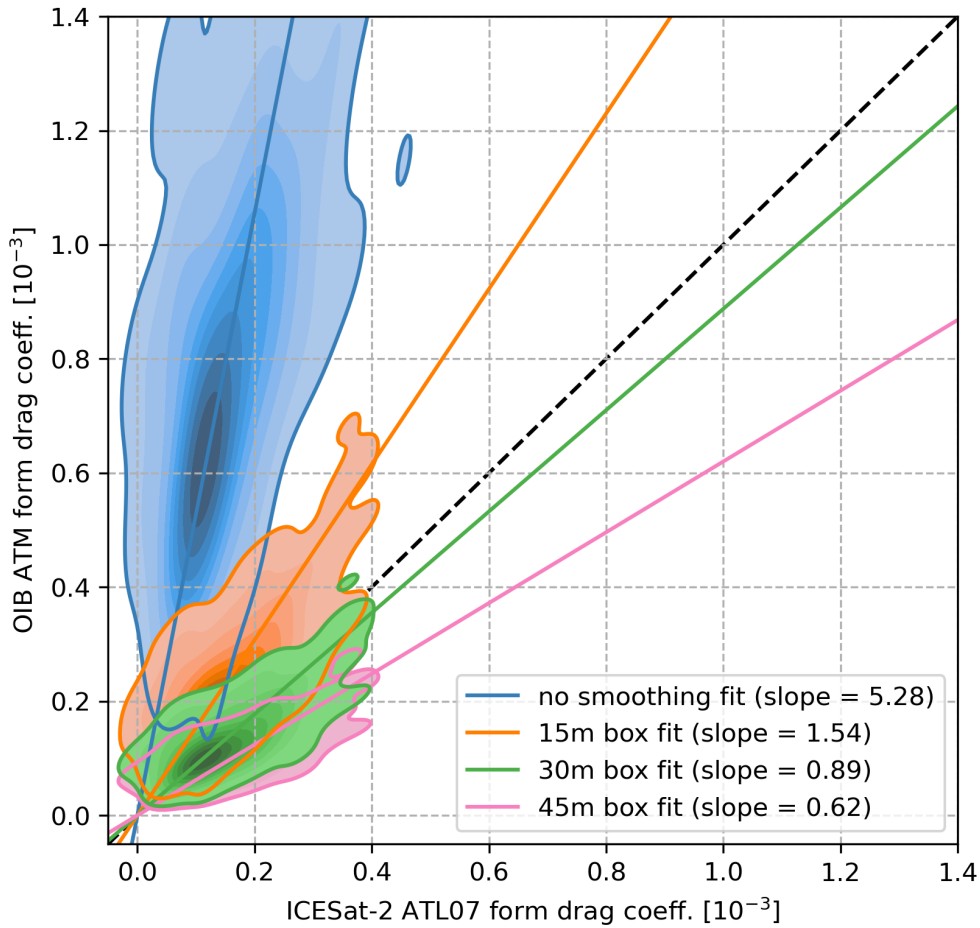

**Figure 2.** Heat maps of 12.5 km grid resampled 10-km average ICESat-2 ATL07 form drag coefficients plotted against those computed from OIB ATM drag coefficients from 4, 8, 19 and 22 April 2019 which are resampled and calculated in the same manner. The blue heat map and line of best fit represent the base drag coefficients from ICESat-2 ATL07 and the full resolution OIB ATM data; this regression is the basis for the scaling applied to ICESat-2 drag coefficients. The other three heat maps feature OIB ATM data smoothed by a moving average filter with a window sizes of 15 m (in orange), 30 m (in green) and 45 m (in pink). The norm for each of the heat maps is different so as to show the full variability of each and avoid oversaturation. The lines represent Huber fits with colour coding matching that of the bivariate heat maps; except for the dashed black line which represents the identity line.

than they really are, this also pushes a lot of small ridges below the cutoff, reducing the sample size. This reduction results in similar averaged values between the smoothed and high-resolution data-sets as can be seen in Figs. 1A and 1B, where the average obstacle height $H_e$ is the same. The only exception are the features that are not detected at all (Ricker et al., 2023),

which force the regression to be steeper than expected. With obstacle spacing, the smoothing gets in the way of extracting any meaningful relationship. As can be seen in Figs. 1A and 1B, the smoothing reduces sample size which is directly proportional obstacle spacing, as less obstacles translates to a higher average spacing between them. It is only through evaluating equation 3 with the input parameters where we see a reasonable relationship. Comparing ICESat-2 ATL07 with OIB ATM form drags with varying box size smoothing applied also shows expected results (Fig. 2), further confirming to us that regressing form

drags is the best approach.

The correlation found between the drag coefficients computed from the different instruments is $0.61$ (blue heat map in Fig. 2), and the mean square error (mse) between the OIB ATM drag coefficients and the ICESat-2 ATL07 coefficients with the scaling factor applied ($5.28$) is $1.1 \cdot 10^{-4}$. Considering some ridges are not detected (Ricker et al., 2023) due to sampling issues, and the lack of perfect coincidence, we do not expect perfect correlation. Moreover, we're looking at spatial averages here, where the

smoothing has a very strong effect on the ridge spacing (as can be seen in Fig. 1), that is why a topography comparison where the sampling of ICESat-2 is simulated with the OIB ATM data, can show better agreement as in Kwok et al. (2019a). However, that is not our aim in this study, here we try to make the Garbrecht et al. (2002) parameterization applicable to ICESat-2 ATL07 data and correct for the sampling issues using OIB ATM. For comparison's sake, we try to simulate ICESat-2 ATL07 with OIB ATM data with the moving average filters in Fig. 2, but we chose not to simulate the elliptical footprint of ICESat-2 in detail

as in Kwok et al. (2019a) and Ricker et al. (2023) for that is not needed for the monthly pan-Arctic drag coefficient product which is the end result of this study. Unsurprisingly, comparing the correlation and mse with the OIB ATM data (in blue) to the smoothed version (30 m box [in green] which has the best agreement with the identity line), we have found a correlation of $0.72$ and a mse of $2.4 \cdot 10^{-6}$ (with the scaling factor $0.89$ as in Fig. 2) for the latter. This better agreement is observed as here the OIB ATM data is sampled similar to how ICESat-2 ATL07 is, making the methods identical will raise the correlation even

higher as in Kwok et al. (2019a). What we require for our study is for the drag coefficients to be calculated as in Castellani et al. (2014) and Petty et al. (2017), making use of high resolution and high sampling of the airborne data-sets, and then regressing the OIB ATM values with estimates of the spatially averaged ICESat-2 drag coefficient. In this way, we aim to improve the ICESat-2 product and amplify the signal that is lower than expected due to sampling.

For an inter-comparison of the drag coefficients processed for each of the three strong beams see Fig. A2 in the appendix. Us-

ing the first and third strong beams we can produce a similar result despite the model being trained with the second strong beam (the most coincident beam). To incorporate the full available high-resolution data-set as well as minimize random sampling errors from here on we use all three strong beams for all ICESat-2 ATL07 parameter maps.

In Fig. 3 we map average obstacle height and spacing used as input in equation 3 as well as the resulting obstacle form drag coefficient ($C_{d,10,o}^n$), with the skin drag coefficient constant ($C_{d,10,s}^n$=$8.38 \times 10^{-4}$) added, for the month of April, 2019.

Here we do not scale with sea ice concentration ($A$) nor consider floe edge and open water drag components so as to focus on the difference between the scaled and base ICESat-2 ATL07 drag coefficients. The areas outlined in Fig. 3A represent the

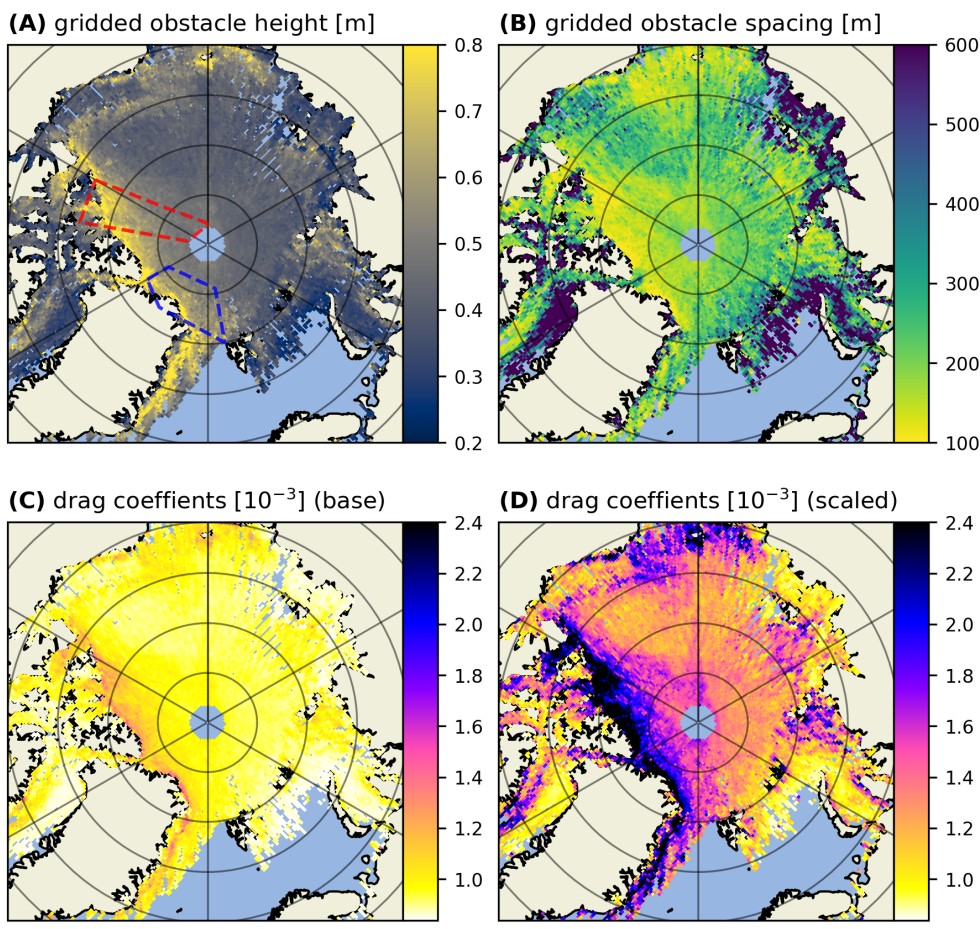

**Figure 3.** Data computed from April 2019 ICESat-2 ATL07 tracks (all three strong beams): (A) average obstacle height, (B) average obstacle spacing, (C) total neutral 10-m atmospheric drag as computed with equation 3 from ICESat-2 average obstacle height and spacing, (D) same as (C) but with the OIB ATM scaling factor (5.28) applied. In (A) zones marked in red and blue represent near-coincident OIB ATM topographic data used to generate the scaling factor via regression (08, 12, 19, 22 April 2019) and data used for evaluation (06, 20 April 2019), respectively.

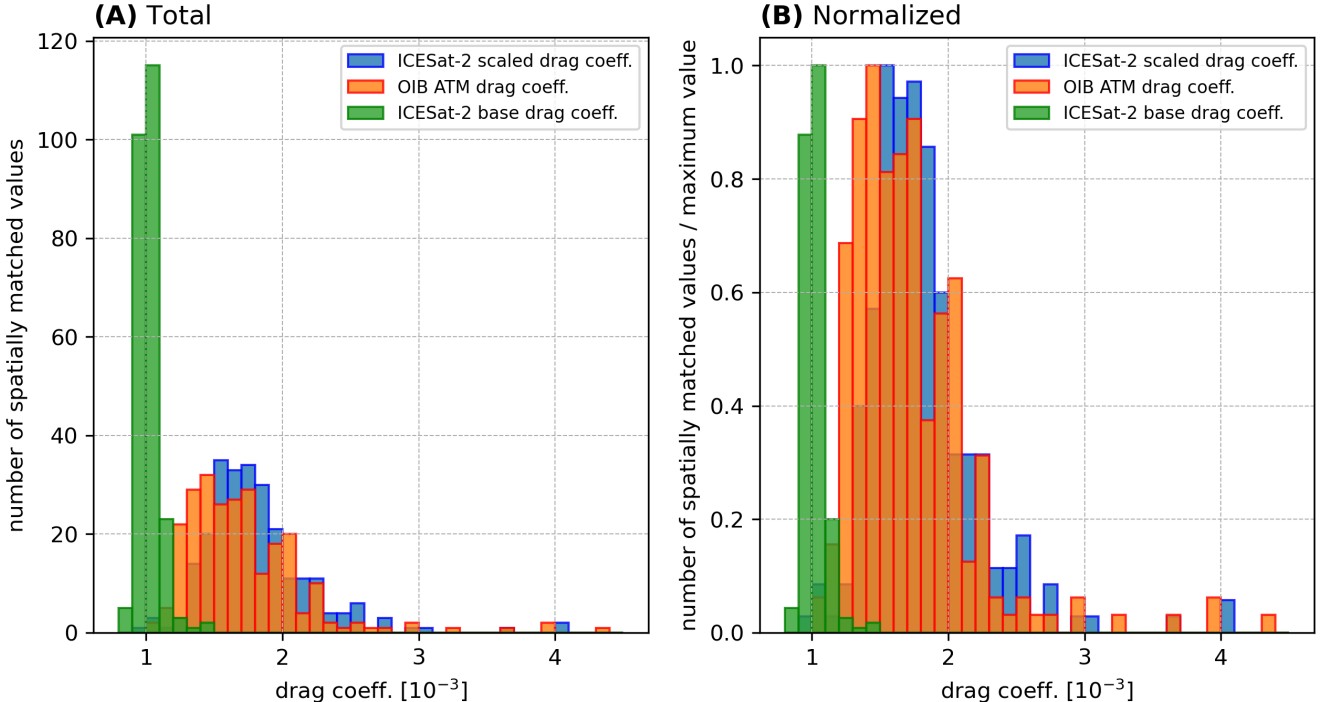

**Figure 4.** Histograms of ICESat-2 ATL07 drag coefficients with (in blue) and without (in green) the scaling factor applied as well as the OIB ATM drag coefficients (in red). Here (A) shows the absolute number of matched grid cells within a given drag coefficient range and (B) is normalized such that every value is divided by the maximum for each data-set.

area where near-coincident OIB ATM flights took place (in red) as well as additional topographic data over sea ice from the month of April 2019 (06.04 and 20.04) that is used for the evaluation study in section. 3.2 (in blue). Looking just at the drag coefficients, in Figs. 3C and 3D, we can see that with the OIB ATM scaling factor applied, the data product is in much better

agreement with the pan-Arctic maps produced in Petty et al. (2017) as well as the regional drag assessments conducted in Castellani et al. (2014). The spatial variability across all parameters in Fig. 3 also confirms the expectation of multiyear ice that is predominantly north of Greenland and the Canadian Archipelago being more rough ($C_d^n > 1.2 \cdot 10^{-3}$ before scaling up; $C_d^n > 2.2 \cdot 10^{-3}$ after) and as a consequence exhibiting a higher concentration of tall ridges ($H_e > 0.8$ m) and thereby shorter spacing ($x_e < 100$ m) between them.

**3.2    Evaluation study**

We take advantage of OIB data from north of Greenland (outlined in blue in Fig. 3A) and collocate it to ICESat-2 ATL07 drag coefficient data produced for the month of April 2019 to perform an evaluation study of our product. In Fig. 4, we compare the drag coefficients computed from the OIB ATM data-set using the methods (see section 2.3) that were used on the near-coincident 'training' data-set (outlined in red in Fig. 3A), to matching grid cells from the 2019 ICESat-2 ATL07 drag

coefficient map. Both the original values (Fig. 3C) in green and those multiplied by the OIB ATM scaling factor (Fig. 3D) in orange are shown along with the ones computed from the OIB ATM data-set. Notably, the distribution of the base drag coefficients is overall much narrower than the other two with the main peak centered around $\sim 1.0 \cdot 10^{-3}$ and a secondary peak at $\sim 1.4 \cdot 10^{-3}$. Meanwhile, the distribution of OIB ATM and scaled ICESat-2 ATL07 drag coefficients both show a similar distribution with the main peaks centered around $\sim 1.6 \cdot 10^{-3}$ and a smaller secondary peaks at $\sim 4.0 \cdot 10^{-3}$. This suggests

that our scaled ICESat-2 ATL07 drag coefficients perform reasonably well to represent the drag variability, at least for this part of the Arctic. Given the two data-sets are retrieved on different days (within the same month) with ice drifting in between, comparing them grid cell to grid cell is not meaningful since drag coefficients vary in time. If further ATM data becomes available from different regions in the Arctic this evaluation should be extended.

### 3.3   Interannual drag coefficient estimates

To increase the temporal coverage of Fig. 3, we look at spatial variability in 3-month aggregates throughout 2019 in Fig 5 (see Figs. A5 and A6 in the appendix for years 2020 and 2021). Here 3 months are chosen to be a reasonable time-frame to maximize the data contained within individual maps on account of ICESat-2's 91-day repeat cycle (e.g., Kwok et al., 2021a, 2019b). All rows of maps within Fig. 5 contain obstacle height, spacing and drag coefficient for consecutive three-month periods.

    In the last column (col. IV), we include the floe edge and open water drag coefficient terms according to equation 5; there

we can observe drag coefficients $> 1.5 \cdot 10^{-3}$ along the marginal ice zone (MIZ). This combined parameterization is our best estimate for satellite derived atmosphere-ice drag. It includes variable form drag due to obstacles and floe edges as well as constants for open water and ice skin drag. However, drag due to floe edges next to over-frozen leads as well as at the edges of melt-ponds in summer is not accounted for (which could be a future enhancement). By looking at the full year separated into 3-month aggregates we can observe the spatio-temporal evolution of drag coefficients Arctic-wide. We observe a seasonal

variability of up to $\pm 1.0 \cdot 10^{-3}$ in some multiyear ice regions though there is a thin band of ice close to the Canadian archipelago that is consistently $> 2.0 \cdot 10^{-3}$. Arctic-wide, this effect is comparatively smaller, but nevertheless a change of up to $\pm 0.5 \cdot 10^{-3}$ in total drag coefficient occurs in most areas of the Arctic. This is consistent for the years 2020 and 2021 as well (see Figs. A5 and A6 in the appendix).

    For both columns III and IV in Figure 5 it is important to mention that the summer months likely exhibit higher levels of

uncertainty, e.g., due to data gaps caused by clouds and due to melt ponds that can saturate the ICESat-2 photon detection system (Tilling et al., 2020). This is a consequence of melt ponds being highly specular and typically reflecting a large amount of signal photons. When ATLAS strong beam timing channels receive more photons than they can handle within a dead time interval, they can no longer detect additional incoming photons; which can lead to short gaps in the topography data. See Tilling et al. (2020) for more information on how ICESat-2 views melt ponds.

To observe the seasonality as well as the monthly evolution of our best estimate of pan-Arctic total neutral drag coefficients on an interannual scale from November 2018 to June 2022, we plot the average drag coefficient, obstacle height and spacing for each month along with the total area of grid cells covered with ICESat-2 data in Fig. 6. Importantly, the total area covered is not the same as sea ice extent and is generally less than the latter due to clouds and returns with ice concentrations $< 15\%$ not

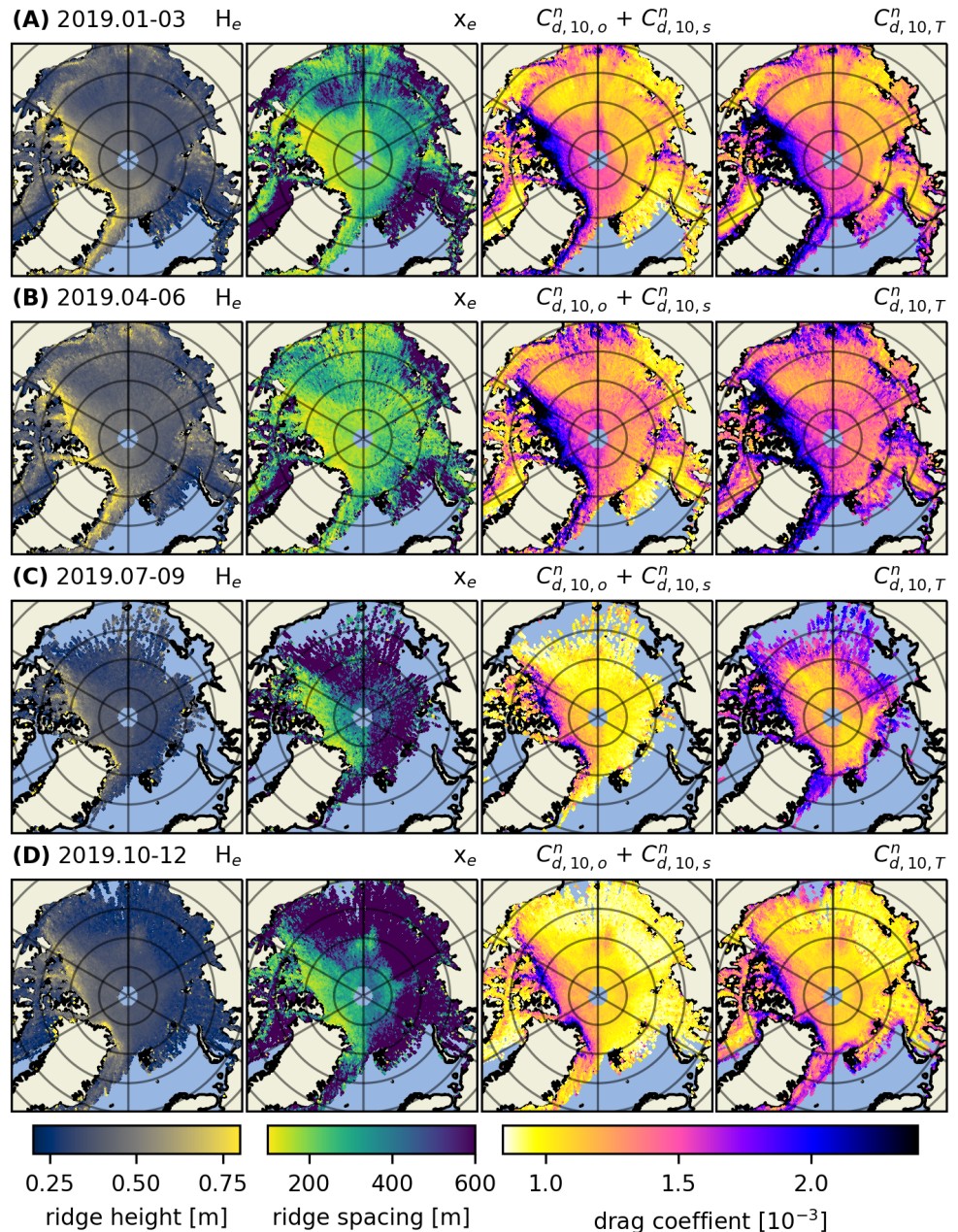

**Figure 5.** 2019 obstacle height ($H_e$), obstacle spacing ($x_e$), drag coefficient as a sum of sea ice skin drag and form drag due to obstacles ($C_{d,10,o}^n + C_{d,10,s}^n$), total drag coefficient as a sum of the sea ice skin drag, form drag due to obstacles and floe edges and open water drag ($C_{d,10,T}^n$). Importantly, columns 1 and 2 are the the obstacle heights and spacings as retrieved from ATL07, whereas in columns 3 and 4 the form drag due to obstacles is multiplied by the OIB ATM scaling factor. The periods for which these parameters are calculated are January to March (A), April to June (B), July to September (C) and October to December (D), 2019.

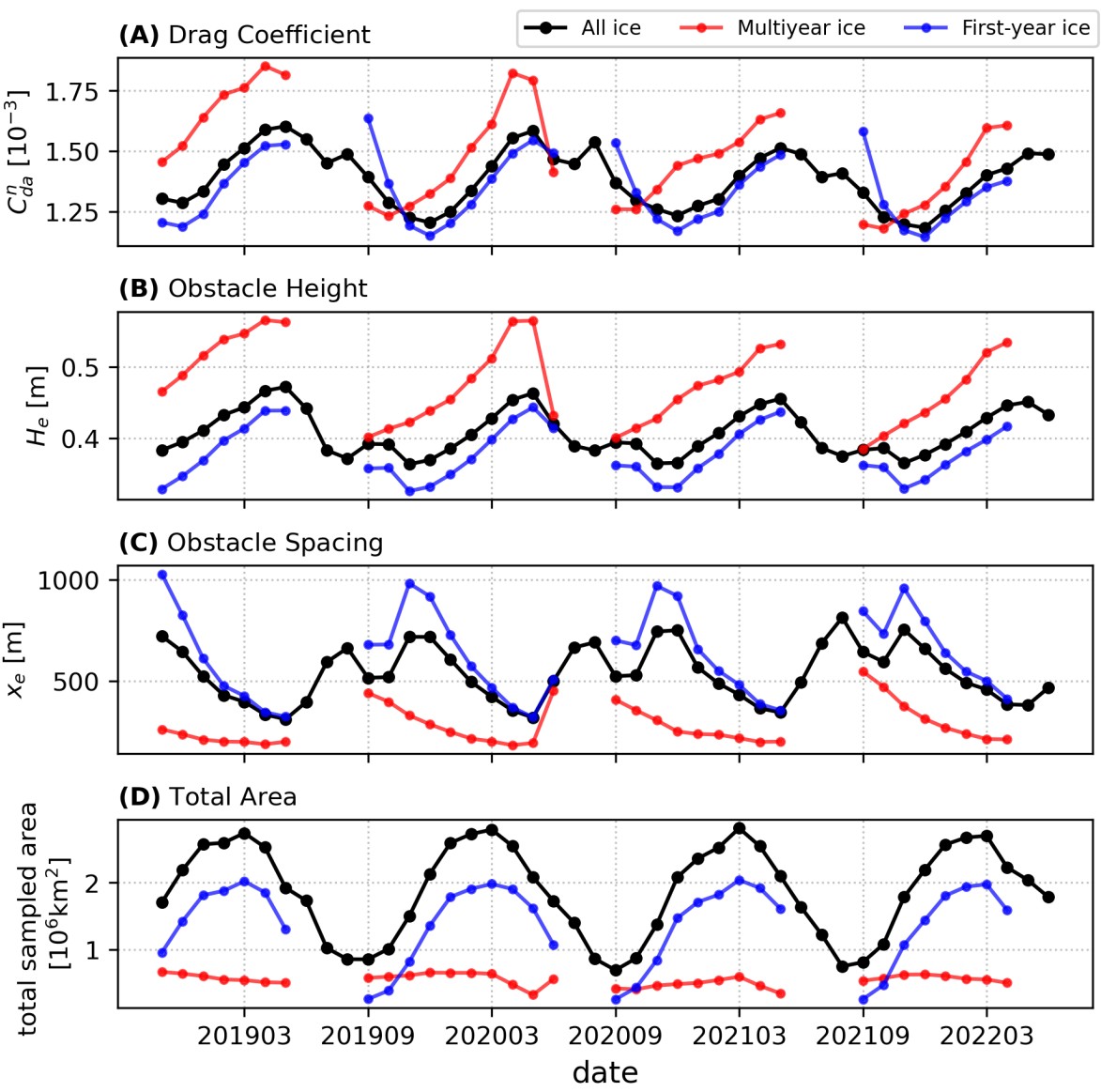

**Figure 6.** Time series from 11.2018 to 06.2022 of total ICESat-2 drag coefficient (as computed in equation 5) with the OIB ATM scaling factor applied (A), obstacle height (B), obstacle spacing (C) and total area covered by ICESat-2 observations (D) for the whole Arctic (in black), multiyear ice (in red) and first-year ice (in blue).

being processed (Kwok et al., 2021b). Notably, both obstacle height and spacing is what is used to calculate the base ICESat-2
ATL07 drag coefficients; for these no corrections are applied and thus it is expected that the heights are underestimated and the
spacing overestimated as compared to OIB ATM due to smoothing by the larger ICESat-2 footprint. In addition to pan-Arctic
averages, we also produce these statistics for multiyear ice (MYI) and first-year ice (FYI). Here, we make use of the MYI
concentration retrieved using brightness temperatures from the microwave radiometer AMSR2 and radar backscatter from the
C-band scatterometer ASCAT (Shokr et al., 2008; Ye et al., 2016a, b; Melsheimer et al., 2022). Sea ice area classified as below
50% MYI according to the retrieval, are considered as FYI and used to compute FYI averages and conversely all values equal
to and above 50% are used to compute MYI averages (see Fig. A7 for the distribution of MYI over 3-month time periods when
available). By comparing the two ice types we can study the differences in their areal averages. As expected we see higher
drag coefficients (MYI: $C_d^n \approx 1.2 - 2.0 \cdot 10^{-3}$, FYI: $C_d^n \approx 1.0 - 1.6 \cdot 10^{-3}$) and obstacle heights (MYI: $H_e \approx 0.4 - 0.6$ m,
FYI: $H_e \approx 0.3 - 0.4$ m), and conversely lower obstacle spacings (MYI: $x_e \approx 200 - 500$ m, FYI: $x_e \approx 250 - 1000$ m) in the
averages from the MYI ice portion of the Arctic. The MYI concentration data product is only available for winter months and
hence the lack of data for the summer months in the time series. As a result this means we are unable to distinguish the FYI
contribution of the form drag due to floe edges peak in August and can only estimate that upper bound given the full data-set.
As for temporal evolution, there is an annual cycle in all three parameters such that the annual maximum (minimum) average
drag coefficient and obstacle height (obstacle spacing) in May lags behind maximum sea ice extent which is typically in March.

**3.4 Spatial and temporal variability**

Looking at our 3-monthly spatial analysis (Figs. 5, A5 and A6) as well as the monthly time series (Fig. 6), we corroborate the
results found in Petty et al. (2017) with the MYI sea ice regions north of the Canadian Archipelago and Greenland exhibiting
high drag ($C_{d,10}^n > 1.5 \cdot 10^{-3}$) and the smooth FYI sea ice regions of the Beaufort (north of Alaska Western Canada), Chukchi
(north of Fram Strait) and Siberian (north of Siberia) Seas exhibiting low drag ($C_{d,10}^n < 1.0 \cdot 10^{-3}$). We corroborate Duncan
and Farrell (2022) in terms of the distribution of spatial variability of 10-km average obstacle spacing, e.g., <200 m near the
Canadian Archipelago, for the winters of 2019, 2020 and 2022 that they have produced using the UMD-RDA algorithm. Based
on the limited amount of data we analysed, we also corroborate that the drag coefficient variability in space is larger than the
variability across seasons as was found by others (Castellani et al., 2014; Tsamados et al., 2014).

We observed interesting features of ice topography, including a tongue of ($C_{d,10}^n > 1.5 \cdot 10^{-3}$) sea ice that forms across the
Beaufort Sea and towards a rough ice patch surrounding the Wrangel Island (near Fram Strait along the antimeridiean) only in
select months (see Figs. 5B and A6B). Similarly, when Arctic sea ice extends across the Arctic Ocean and to Siberia, Severnaya
Zemlya is often (but not always) surrounded by rough ice as well ($C_{d,10}^n > 1.5 \cdot 10^{-3}$) (see Figs. 5B and A5B). These effects
may be attributed to the movement of the Beaufort Gyre as well as to the tendency of ice to ridge near land, respectively.
Notably, within the time span we analysed, May is the month that repeatedly exhibits annual minimum obstacle spacing and
annual maximum obstacle height and drag coefficient. This supports the notion that sea ice–atmosphere drag exhibits an annual
cycle (e.g., Andreas et al., 2010). By also including drag due to floe edges we also observe a smaller peak in August, when
the ice-water boundary is at its longest. We observe a decrease in the yearly maximum average drag coefficient across all ice

types during the four years we looked at, but given the short time-frame we cannot attribute this decrease to anything more than natural variability.

### 3.5 Uncertainty due to sampling

While ICESat-2 has a very high resolution when compared with other laser altimeter satellites, it's still larger than the 1 m resolution of the OIB ATM data. The ATL07 segment length of about 30 m, over which 150 signal photons are obtained to lower noise in the height retrieval, smooths out the topography via the dual-Gaussian fit much like the moving average filter we applied to the OIB ATM data. This smoothing effect is discussed in detail in a recent study by Ricker et al. (2023) where coincident ICESat-2 ATL07 and airborne Altimeter Laser Scanner (ALS) data from the MOSAiC (Multidisciplinary drifting Observatory for the Study of Arctic Climate) expedition were compared. Ricker et al. (2023) show that the ICESat-2 ATL07 strong beam could detect only 16% of obstacles above the threshold of 0.6 m that were registered by ALS. A comparatively higher detection rate of 42% was achieved by processing ATL03 by using a higher-resolution topography data-set (Duncan and Farrell, 2022). Notably, neither of the two ICESat-2 sea ice height products were able retrieve the full extent of surface topography (Ricker et al., 2023). Assuming the lower threshold value of 0.2 m used in this study, we can expect these detection rates to rise but at some point hit a limit imposed by ICESat-2 ATL03's footprint of 11 m (Magruder et al., 2020, 2021) that is inferior to the resolution used in most modern airborne surveys looking at topography, e.g., OIB ATM, ALS. Thus for the purposes of our pan-Arctic study, we have chosen to stick with the publicly available and regularly updated ATL07 data-set as either of these two data products will require some type of correction if realistic drag coefficient estimates are to be computed from them. While ATL07 has a lower obstacle detection rate locally and the obstacle height (spacing) is typically overestimated (underestimated), the spatial information on Arctic-wide obstacle distribution should be conserved according to our comparisons to airborne data (Section 3.1). That is why we use a regression transfer model that is trained by near-coincident OIB ATM to scale up these underestimated ICESat-2 drag values and obtain them closer to the expected form drag range estimated from higher-resolution airborne laser data.

How representative the scaling factor is for the whole of the Arctic is difficult to gauge and with limited spatial and temporal near-coincident coverage we expect there to be some uncertainty. Despite these limitations, the racetrack OIB flights from 8 and 12 April 2019 were flown over two distinct ice types. The 8 April racetrack was 100 km north of the Sverdrup Islands (80.5°N) and the 12 April one was centered at 86.5°N in the central Arctic (Kwok et al., 2019a). As a result, the former was over thicker and rougher ice, while the latter was over thinner and smoother ice giving us the opportunity to see how the drag coefficients compare between the two instruments in the different regimes. The scaling factors derived for the two different days are 4.42 and 5.36 respectively, resulting in an uncertainty that is in the range of $\pm 17.5\%$. This small discrepancy can also be explained by ATL07 sampling: with a smaller obstacle frequency over smooth ice the likelihood of not detecting the few that are present increases (Ricker et al., 2023) thereby increasing the obstacle spacing used in the calculation of drag coefficients for every 10 km segment. Where the obstacle density is generally high, like in rough deformed areas near the Canadian Archipelago, though the detection rate may be low, there will always be an ample amount per 10 km segment to detect a higher drag coefficient signal. Thus, the sampling issue with regard to computing drag coefficients from topography

features is more prevalent over smooth ice than rough ice and a higher correction is needed. As 19 and 22 April OIB flights cover larger areas and the rougher deformed ice near the Archipelago is rather small in extent, the scaling factor derived from all 4 days is closer to that of the April 12 racetrack and more representative for the whole Arctic that is predominantly smoother than the ice surveyed on the 8th of April.

## 3.6 Discussion and concluding remarks

In this study we used a combination of the Garbrecht et al. (2002) and Lüpkes et al. (2012) parameterizations to calculate obstacle drag coefficients. Although it is important to understand that this method has some uncertainty, it represents the state-of-the-art. Currently, it is the only available parametrization of drag coefficients accounting for the effect of pressure ridges over closed sea ice cover and of floe edges in regions with fractional sea ice cover. In the following, we address the background and the uncertainties of the parameterizations, e.g. on the basis of results obtained with alternative formulations.

The parameterization idea for the effect of ridges (and other sea ice features over closed sea ice) has been first tested by Garbrecht et al. (1999) on the basis of turbulence measurements made at the bow mast of RV Polarstern when the ship was drifting at different positions downstream of a large pressure ridge. Thus, this data set was independent from the airborne turbulence data which were the main data source for Garbrecht et al. (2002). The latter data-set was validated once more in a thesis by Ropers (2013), who compared the Garbrecht et al. (2002) drag coefficients with drag coefficients derived from additional airborne turbulence and topography data. Furthermore, Castellani et al. (2014) showed that at least the average neutral 10-m drag coefficient, obtained from the Garbrecht et al. (2002) parametrization with parameters given in Section 2.4, agrees well with values for closed sea ice derived from Andreas et al. (2010) using SHEBA data.

It is important to understand that all available sea ice form drag parameterizations including those for the effect of pressure ridges and of floe edges are based on a similar formulation of dynamic pressure acting on these obstacles (Gryanik and Lüpkes, 2023). While the Lüpkes et al. (2012) approach, first formulated in a modified version by Hanssen-Bauer and Gjessing (1988), is a 2D approach, the Garbrecht et al. (2002) approach is only 1D. Ropers (2013) investigated if more complex assumptions for the latter scheme concerning the ridge geometry would improve the results, but the main conclusion was that more complex models require input variables, which are usually not available. It is furthermore important that the addition of ridge form drag to the scheme of Lüpkes et al. (2012) considered here does not represent a competing scheme. On the contrary, the formulation of Lüpkes et al. (2012) allowed the specification of the ridge component from the very beginning. Only for simplicity, they assumed an average roughness of ridge-covered sea ice to concentrate their work just on the edge effect. The latter was expected to be dominating in the MIZ and perhaps also in the inner Arctic under melt conditions with many leads and ponds.

The approach for floe edge form drag was also used in mesoscale modelling studies. Vihma et al. (2003) showed that the application of the scheme led to a very good agreement of modelled and observed meteorological mean variables and turbulent fluxes. Inclusion of form drag in the marginal sea ice zones using the Lüpkes et al. (2012) scheme with parameter values based on Elvidge et al. (2016) resulted in an improvement of atmospheric model results (Renfrew et al., 2019). Birnbaum and Lüpkes (2002) also investigated the effect of floe-edge generated form drag in the marginal sea ice zone on meteorological parameters by applying a mesoscale nonhydrostatic model. They pointed to the importance of a proper choice of the coefficient

of resistance $c_w$ to obtain realistic fluxes when the form drag parameterization was included. Finally, Martin et al. (2016) show that the inclusion of atmospheric form drag leads to improvements in the modelling of sea ice drift. The latter work addresses only floe edge form drag but one can expect that further improvement is possible when ridge-generated form drag is included as well.

Thus, for the analysis of uncertainty we concentrate on the combined approach as described in section 2.4. Naturally, the chosen values and formulations for, e.g. the coefficient of resistance, value of the skin drag coefficient and the inclusion of the sheltering function contain their own uncertainties, especially when generalizing in time and space. As mentioned in Section 2.4, for the coefficient of resistance $c_w$, here we use an approach by Garbrecht et al. (2002) where $c_w$ depends linearly on the obstacle height $H_e$. The given coefficients in this parameterization $0.185$ and $0.147$ have some uncertainty because they have been derived from pressure measurements over only a few sea ice ridges. For this reason, we performed sensitivity studies with different $c_w$ formulations (e.g., Garbrecht et al., 1999; Ropers, 2013). Results are shown in Fig. A4 for April 2019 where the obstacle form drag coefficients have been calculated with the different formulations for the coefficient of resistance, taking into account necessary adjustments (modified aerodynamic roughness length and thus adjusted neutral skin drag coefficient, e.g. in the case of the Ropers (2013) version). The conducted studies showed that the principal results (geographic distribution) were unchanged but small differences between drag coefficients are observed with the different formulations of the coefficient of resistance $c_w$. The standard deviation (mean) was found to be $2.5 \cdot 10^{-4}$ ($2.9 \cdot 10^{-4}$), $4.6 \cdot 10^{-4}$ ($5.3 \cdot 10^{-4}$) and $5.8 \cdot 10^{-4}$ ($6.1 \cdot 10^{-4}$) for the Garbrecht et al. (2002) formulation with the original coefficients ($c_w = 0.05 + 0.14 H_e$), the version with the natural logarithm ($c_w = 0.22 \ln(H_e/0.2)$) and the Ropers (2013) formulation ($c_w = 0.05 + 0.35 H_e$), respectively. These results are altogether not too different from the Garbrecht et al. (2002) formulation using $c_w = 0.185 + 0.147 H_e$). The standard deviation (mean) amounts to $4.6 \cdot 10^{-4}$ ($6.0 \cdot 10^{-4}$). Most importantly, the spatial distribution of high and low obstacle form drag regimes is conserved independent of the used $c_w$ parameterization.

Here, we tested a different hierarchy level of the Lüpkes et al. (2012) scheme than that which is used for this study (level 4). It is their level 2 parameterization which allows for specifying the measured grid-cell averaged freeboard. Instead of the constant value $0.41$ m that is implicitly used in the Lüpkes et al. (2012) version used in the previous sections, we considered the data from ATLAS/ICESat-2 L3B Daily and Monthly Gridded Sea Ice Freeboard, Version 3 (ATL20) thereby implementing freeboard from satellite remote sensing measurements. Because of the smoothing imposed by sampling the results did not show any significant improvement over using constant freeboard $h_f = 0.41$ m as recommended in Lüpkes et al. (2012) for the simpler level. Ideally, all components of floe edge form drag coefficients should be taken from remote sensing to better monitor the changing Arctic system, but especially with regards to floe edge sizes, ICESat-2 cannot reliably determine this parameter Arctic-wide. Though it is beyond the scope of this study, we encourage future work in this direction with a multi-satellite approach that might remedy the limitations of each individual instrument. However, since in the Lüpkes et al. (2012) parameterization form drag still depends on the sea ice fraction $A$, the contribution of the form drag is rather small in regions with $A$ near 1. This is the main reason why form drag is still underestimated by the level 2 scheme even when variable freeboard is allowed. This holds especially near Greenland where ridges are tall. With respect to the application discussed here, the Lüpkes et al. (2012) floe edge form drag parameterization has uncertainties imposed by the limitations of satellite remote

sensing. Namely, here we use ice concentration derived from passive microwave data (AMSR2 microwave radiometer in this case) as input, overfrozen leads and ponds are not considered. As this freezing can happen already in August, the overall drag from floe edges is likely underestimated then. The proportionality coefficient $3.67 \cdot 10^{-3}$ of the Lüpkes et al. (2012) level 4 hierarchy parameterization, carries with it its own uncertainty. It depends, e.g. on floe sizes and sea ice freeboard (see Lüpkes et al. (2012)). Their uncertainty stems from the fact that the constant is region dependent. For this reason, sensitivity studies were carried out (not shown), in which these values were varied within realistic and recommended ranges given in Lüpkes et al. (2012) (see also Elvidge et al. (2016) and Srivastava et al. (2022)). The result of this sensitivity study was that since the effect of ridge form drag was found to be much larger than the floe edge form drag, the variability of the above-mentioned proportionality constant had only small impact on the total drag coefficient.

The Garbrecht et al. (2002) approach contains in principle also the effect of sheltering by ridges. However, the sheltering function $(1 - \exp(-0.5 x_e / H_e))^2$ (e.g., Hanssen-Bauer and Gjessing, 1988; Lüpkes et al., 2012; Castellani et al., 2014), discussed in section 2.4, was not applied to the main ICESat-2 ATL07 data after it was tested to see if it produced a non-negligible signal. By comparing the month of Arpil 2019 (shown in Fig. 5), with and without the sheltering function implemented, the averaged absolute difference for all filled 25 km grid cells was $7.39 \cdot 10^{-12}$. Thus, such a negligible difference further confirmed that the use of the sheltering function for ICESat-2 ATL07 data was not significant.

Our value for the skin drag differs from the one used in Lüpkes et al. (2012), since there, the effects of ridges and other obstacles are included in the skin drag coefficient as mentioned. By using the smaller skin drag coefficient and a variable obstacle form drag coefficient (e.g., Castellani et al., 2014; Petty et al., 2017), we may introduce a more realistic obstacle form drag, since, as has been shown in this study, it varies a lot in time and space; whereas skin drag over smooth ice we can assume to be relatively constant in comparison. In the combined total drag (as derived in equation 5), the Garbrecht et al. (2002) obstacle form drag and Lüpkes et al. (2012) floe edge form drag parameterizations are meant to be used together to better assess pan-Arctic drag coefficients.

As mentioned previously in section 1, the drag coefficient $C_d$ also depends on the surface roughness dependent stability function $f_m$, for which numerous versions exist (see e.g. Gryanik and Lüpkes (2018, 2023). For this study we have limited our research to assessing the neutral drag coefficients $C_d^n$. In case of stable stratification $C_d$ becomes smaller than $C_d^n$, whereas unstable stratification with more turbulence causes $C_d$ to be greater than $C_d^n$ (Lüpkes and Gryanik, 2015). The local-near surface stratification is heavily impacted by open-water that facilitates upward heat fluxes (Andreas and Cash, 1999; Lüpkes and Gryanik, 2015) and as a result varies between the more ice-covered inner Arctic and the MIZ where open water is more common. Thus, it is in summer, where more open water is present across the Arctic ice cap, that our estimates of the neutral drag coefficients $C_d^n$ are likely below $C_d$. Conversely, over regions with large sea ice cover, the stratification is expected to be more stable in winter during polar nights (Lüpkes and Gryanik, 2015), which will act to offset the impact of higher form drag, suggesting our estimates of $C_d^n$ for winter are more representative of $C_d$.

### 3.7 Significance and novelty of the analysis

Using our best estimates, we have demonstrated that drag force between Arctic sea ice and the atmosphere varies annually throughout the year (see Fig. 6). The implication of this finding is that the turbulent surface flux of momentum, given in equation 1, varies also. In other words, depending on the month of the year, the ice is either more or less susceptible to movement depending on the amount of energy transferred to it via the atmosphere, and by extension, the ocean. We include the ocean here because the sources of atmospheric drag we looked at, primarily form drag due to obstacles, are closely related to the magnitude of oceanic form drag on account of pressure ridges having both a sail (the part above water) and a keel (the part below water) in roughly the same location (Timco and Burden, 1997; Tsamados et al., 2014). Similarly, form drag due to floe edges is also subject to energy transfer from the ocean for the majority of the ice edge that is below the water level. Thus both oceanic and atmospheric form drag are expected to be both temporally and spatially correlated to one another, wherein the oceanic drag is higher in magnitude (Tsamados et al., 2014). Form drag from meltpond edges, a parameter we did not look at here, we expect to be a unique component of total atmospheric drag.

We observe that MYI ice exhibits highest drag in May (red line in Fig. 6A), due to an increase in the form drag due to obstacles, and FYI ice peaks sometime in July-August (according to the secondary peaks in the black line [all data] in Fig. 6A and the associated presumed trajectory of the blue line [FYI data]) from a longer ice-water edge and the associated floe edge drag in summer months. Looking at the gridded data (Figs. 5, A5 and A6), we can further comment on developments on regional scales. Notably it is the Lincoln Sea, north of Greenland, which exhibits the highest form drag due to obstacles with high drag coefficients (2-3x higher than smooth FYI areas, e.g. the East Siberian Sea) reaching as far north as 85 deg in the months of spring (rows A and B). However this is not consistent throughout the year as these relatively high drag coefficients tend to retreat towards the Canadian archipelago throughout summer and autumn (rows C and D). Interestingly, it is not consistent across all years either as this behaviour was not observed in 2021 (Fig. A6). Similarly, the neighbouring Beaufort Sea and Fram Strait (mixture of MYI and FYI) also exhibit wide areas of higher form drag coefficients sometime in late spring (row B). All other Arctic Seas (mostly FYI) primarily show an increase in form drag due to floe edges along the MIZ (see column IV), but also in small part higher form drag due to obstacles near land features. Thus, this data proves highly valuable in terms of identifying previously unknown spatial and temporal developments in pan-Arctic and regional drag. This analysis is the first of its kind as previous studies either assumed uniform drag across the Arctic or did not provide sub-yearly temporal information.

In terms of climate modelling, our findings show that assuming a constant drag coefficient in both space and time misrepresents the variability of momentum fluxes near the surface and thus the main forcing of sea ice drift. This misrepresentation might cause in turn many other deficiencies in air ice interaction such as insufficient variability in the sea ice concentration. Accordingly, a suitable further development of drag parameterizations for a more realistic representation of form drag seems necessary. As for understanding Arctic sea ice, we believe this data has the potential to help with better understanding the interaction between sea ice, ocean and atmosphere, better predict the motion of sea ice and identify temporal and spatial variability of pan-Arctic drag coefficients on a monthly basis. Most importantly, this study helps us link yet another crucial sea ice parameter to remote sensing. This link, given ICESat-2 or similar future mission data is available for years to come, has

the potential to help us better understand the multiannual changes in Arctic sea ice cover as the local climate warms at an unprecedented pace (e.g., Serreze and Barry, 2011; Stroeve et al., 2012).

## 4 Summary and outlook

This study makes use of measured sea-ice topography to calculate atmospheric drag coefficients across the Arctic ice cap on monthly and 3-monthly temporal scales. To our knowledge, it is the first analysis of monthly pan-Arctic drag coefficient estimates of its kind. The sea-ice topography is obtained from the ICESat-2 ATL07 data product at variable resolutions that depend on surface reflectivity but average around 30 m for the strong beams (Kwok et al., 2019b). Using methods developed in Garbrecht et al. (1999) and Garbrecht et al. (2002) according to the drag partitioning scheme proposed by Arya (1973, 1975),

we obtain obstacle, i.e. ridges, height and spacing averages for 10-km segments. We then combine the estimated form drag due to obstacles with sea ice skin drag, drag due to floe edges and a drag due to open water; all of which are incorporated as constants scaled differently with sea ice concentration.

In conclusion, from our analysis of pan-Arctic drag coefficients from the year 2019 and to a lesser extent 2018, 2020, 2021 and 2022, we have observed several noteworthy natural phenomena. Pan-Arctic form drag due to obstacles follows an annual

cycle that is similar in both MYI and FYI regions. The yearly maximum average drag coefficient is not connected to the yearly maximum sea ice extent and seems to occur after the sea ice extent maximum. Form drag due to obstacles is primarily spatially variable (high in MYI regions and low in FYI regions) but nevertheless shows some temporal variability (Maximum in May and minimum in December). Our results suggest that form drag due to floe edges is more prevalent during summer months when large areas are broken up and the MIZ expands, whereas form drag due to surface features peaks in late spring when its

contribution is magnified from MYI regions north of the Canadian Archipelago and Greenland.

While it is beyond the scope this study, we propose the possibility of extending ICESat-2-based analysis to also estimate form drag due to floe edges from satellite measurements rather than using a constant as mentioned in section 3.6. We encourage the open water drag component to be derived from a parameterization that takes into account wind speed and therefore wave height that might cause additional form drag across water surfaces. We propose the use of lead and meltpond data to account

for additional sources of drag not included in our study, e.g. lead and meltpond edges.

*Data availability.* The AMSR2 ASI Sea Ice Algorithm sea ice concentration data product Melsheimer and Spreen (2019); Spreen et al. (2008) and AMSR2 / ASCAT Multiyear ice concentration data product Shokr et al. (2008); Ye et al. (2016a, b); Melsheimer et al. (2022) are used from local repositories and publicly available at https://seaice.uni-bremen.de/data-archive/.

NASA ICESat-2 ATL07 sea ice height (version 5) data product is obtained from https://doi.org/10.5067/ATLAS/ATL07.005, (Kwok et al.,

2021b). NASA OIB ATM L1B elevation data product (version 2) is obtained from https://doi.org/10.5067/19SIM5TXKPGT, (Studinger, 2013).

The processed data files used to produce the figures in this research can be found on Pangaea (Mchedlishvili et al., 2022).

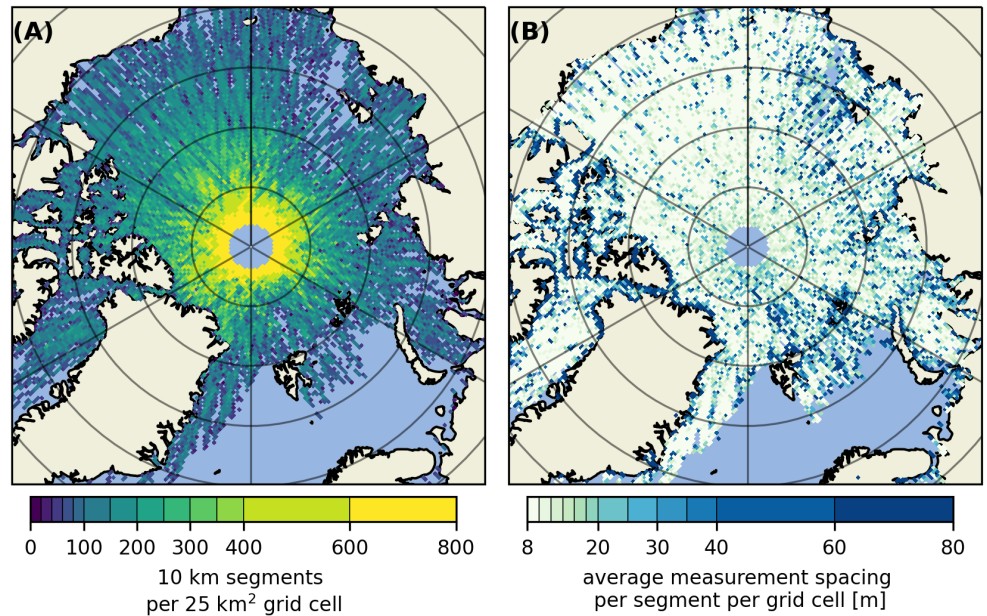

**Figure A1.** The data distribution for the 04.2019 drag coefficient map given as the number of 10 km segments from all strong beams per 25 km² grid cell (A). The average point spacing within each 10 km segment per 25 km² grid cell (B).

## Appendix A: Supporting Figures

Using our methods, we obtain a sufficient amount of data to mostly fill a polar-stereographic 25 km grid via bucket resampling for each month to produce a pan-Arctic monthly total neutral atmospheric drag coefficient analysis. On account of ICESat-2's near-polar orbit the data density is highest around the pole hole and wanes at lower latitudes (see Fig. A1A). As a result, the regional drag coefficient estimates at higher latitudes are more representative of the time periods shown in Figs. 3, 5, A5 and A6, whereas those at lower latitudes are computed with fewer height measurements (often just a few select days). In other words, rather than a temporal mean of surface topography, it is a data set that is sewn together with the best representation of the temporal mean near the pole hole. However, mind that we do not see any discontinuities due to variable sampling in the final atmosphere-ice drag maps. In Fig. A1B one can observe the typical spacing between ATL07 height estimates, which is typically around around ~11-13 m but can be higher due to dark surfaces over which up to 200 m might be needed to collect the sufficient signal photons Kwok et al. (2021a). Similarly, clouds can also increase the spacing as no measurements are retrieved beneath them.

For a comparison between different beams, all of which we combine in our final data product, we refer the reader to Fig. A2. Inter-beam variability due to different range biases is present and was reported on by the ICESat-2 Project Science Office (PSO) in their preliminary analysis (e.g., Bagnardi et al., 2021). In addition, there is the 3.3 km inter-beam spacing which suggests ridges and snow features captured by one beam might not be captured by the rest. At first look Fig. A2D, the inter-

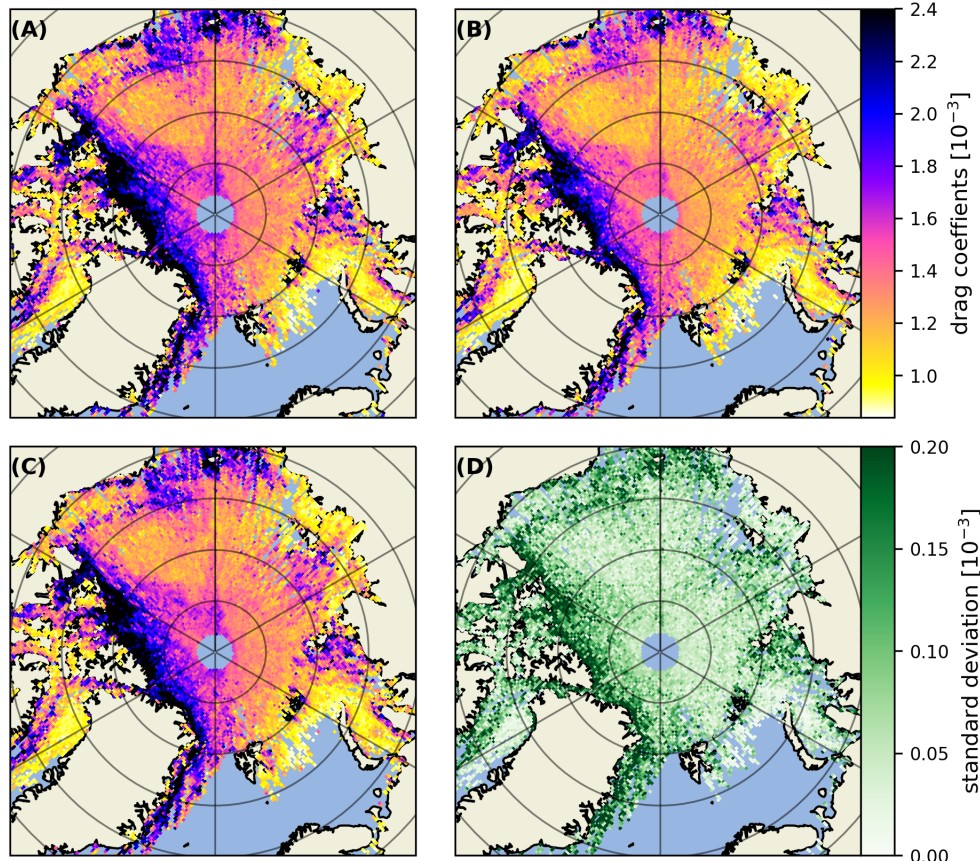

**Figure A2.** Comparison between drag coefficient estimates (sea ice form drag + skin drag) computed from the 1st (A) 2nd (B) and 3rd (C) strong beams as well as the standard deviation between them (D). All three examples have the OIB ATM scaling factor applied.

beam standard deviation, suggests more variability in the MYI rough ice areas but that is because the OIB ATM scaling factor applied to all data scales up all drag coefficients linearly, and hence the variability is increased in those areas as well.

Though it is not the subject of this study, we also briefly looked at the relation between the parameters extracted from ICESat-2 ATL07 which were used in equation 3, with respect to each other as well as the form drag due to obstacles derived from them. We corroborate Brenner et al. (2021), who looked at the keels instead of sails of ridges, that indeed obstacle height and spacing exhibit a negative correlation. Though not always associated (Tin et al., 2003), sails and keels are predominantly spatially coincident and are therefore expected to exhibit proportional heights and depths and similar spacing. When looking at the non-linear cutoff at 200 m for ridge spacing that can be seen both in Fig. A3B and Fig. A3C, it is important to once again consider the "smoothing" and low obstacle detection rates (Ricker et al., 2023) of ICESat-2 ATL07, that are likely the cause of average obstacle spacing not being any lower than what is observed.

Here (Fig. A4) we also show the sensitivity studies done with different coefficient of resistance $c_w$ formulations.

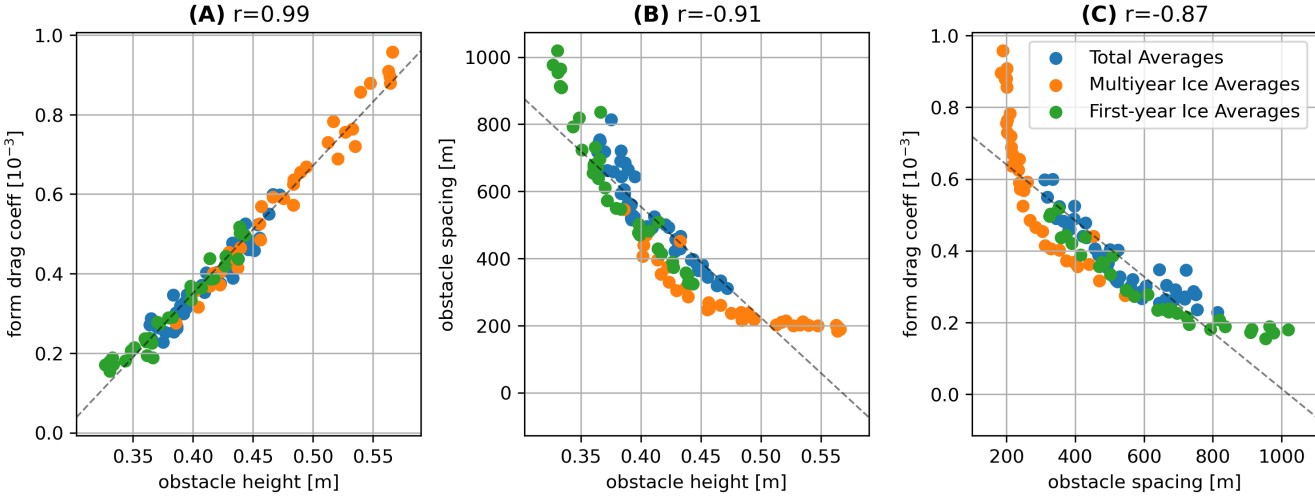

**Figure A3.** Scatter plots between (A) obstacle height and form drag coefficient, (B) obstacle height and obstacle spacing and (C) obstacle spacing and form drag coefficient. All values are taken from Fig. 6, such that blue dots represent the monthly pan-Arctic averages, orange dots represent monthly MYI averages and green dots represent monthly FYI averages.

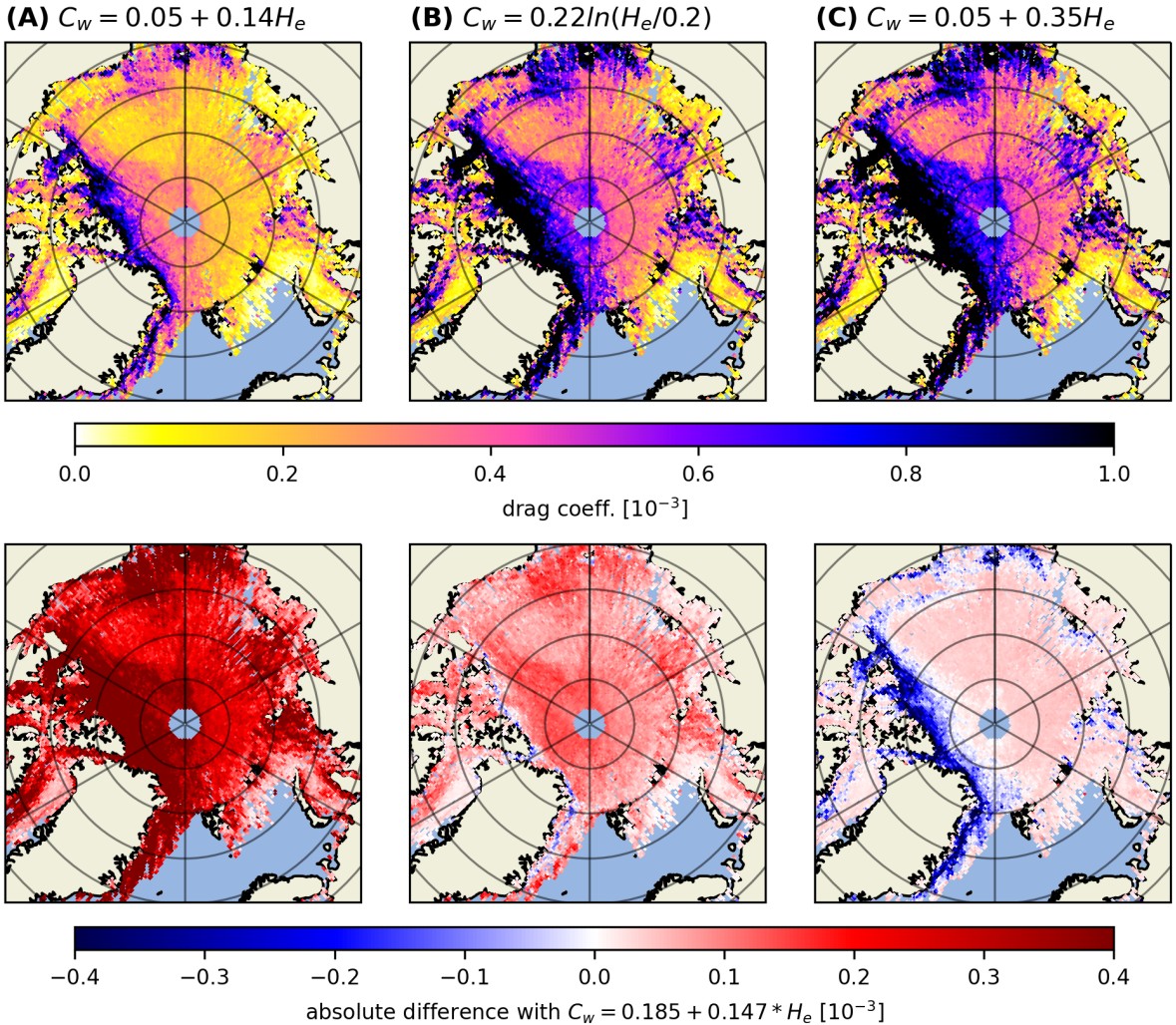

**Figure A4.** Obstacle form drag coefficient monthly maps for April 2019 subdivided into columns labelled by the coefficient of resistance formulation used. (A) uses $c_w = 0.05 + 0.14 H_e$ (Banke and Smith, 1975; Garbrecht et al., 2002), (B) uses $c_w = 0.22\ln(H_e/0.2)$ as suggested by Garbrecht et al. (1999) with all $H_e$ values below 0.5 set to 0.2 to avoid very low and negative values, and (C) uses $c_w = 0.05 + 0.35 H_e$ from Ropers (2013) with an adjusted aerodynamic roughness length of $z_0 = 10 \cdot 10^{-7}$ m. The second row shows the absolute difference between drag coefficients for each of these $c_w$ formulations as compared to the one used in this study: $c_w = 0.185 + 0.147 H_e$ with the modified coefficients from Garbrecht et al. (2002).)

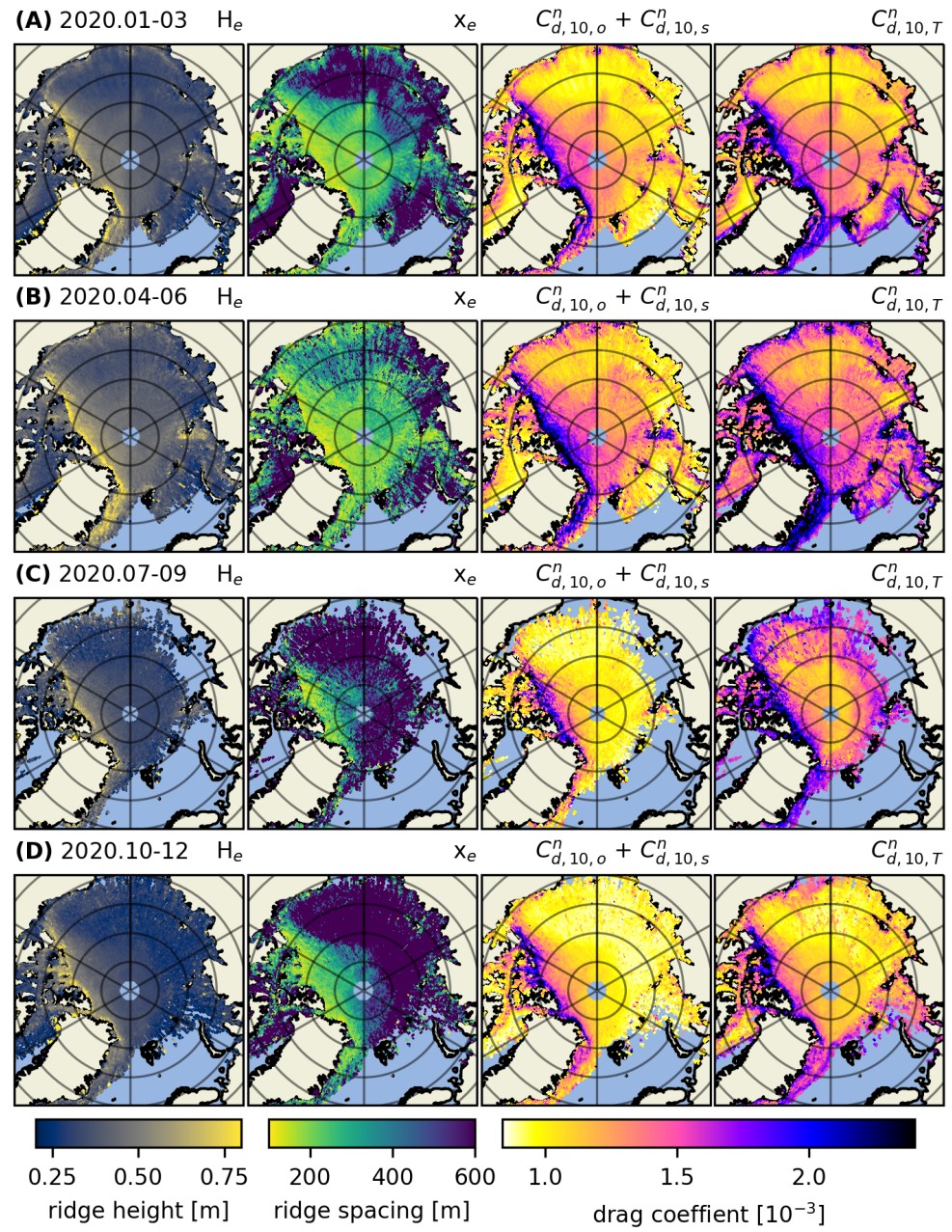

**Figure A5.** Same as Figure 5 but for 2020. Obstacle spacing ($x_e$), drag coefficient as a sum of sea ice skin drag and form drag due to obstacles ($C_{d,10,o}^{n} + C_{d,10,s}^{n}$), total drag coefficient as a sum of the sea ice skin drag, form drag due to obstacles and floe edges and open water drag ($C_{d,10,T}^{n}$)

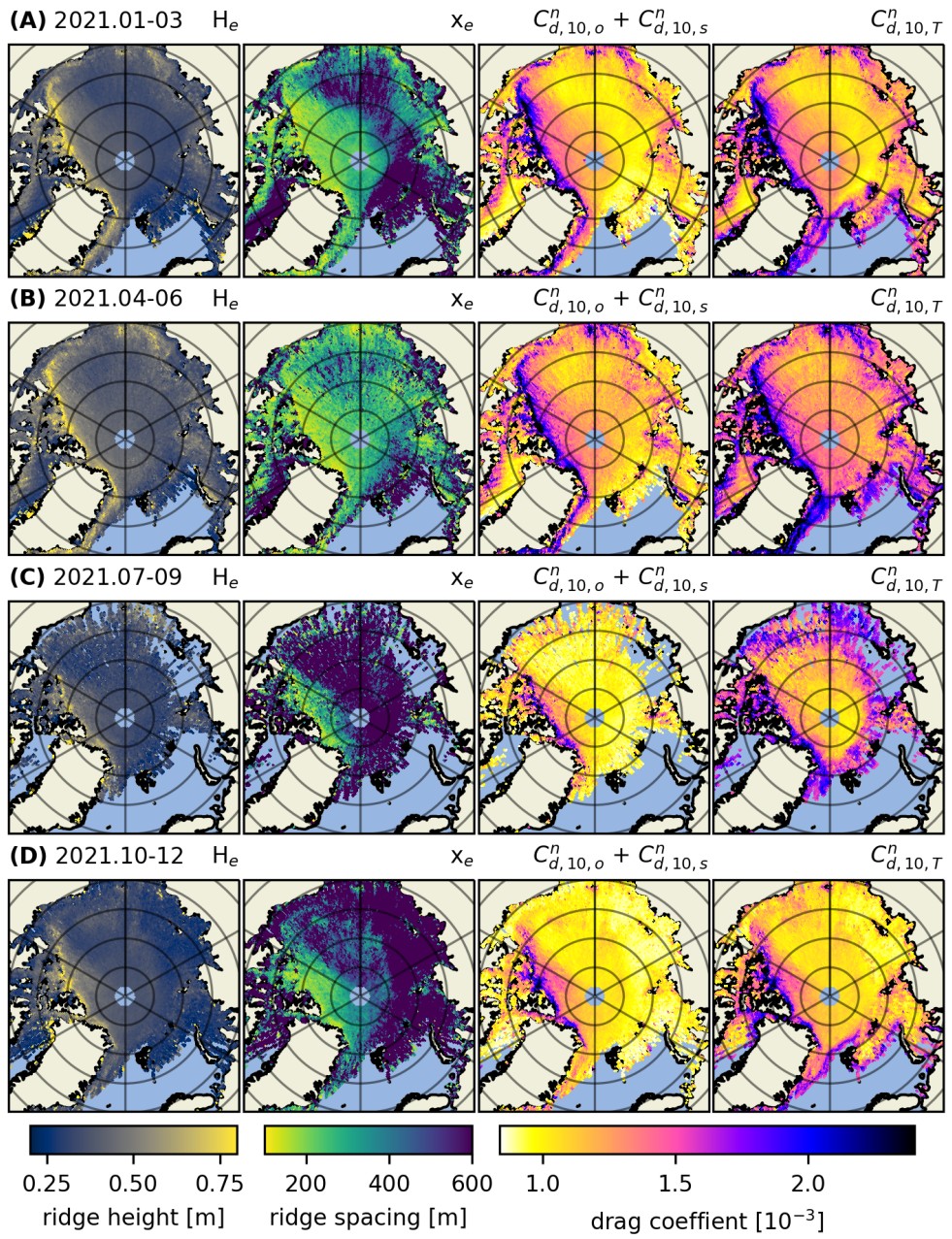

**Figure A6.** Same as Figure 5 but for 2021. Obstacle spacing ($x_e$), drag coefficient as a sum of sea ice skin drag and form drag due to obstacles ($C_{d,10,o}^n + C_{d,10,s}^n$), total drag coefficient as a sum of the sea ice skin drag, form drag due to obstacles and floe edges and open water drag ($C_{d,10,T}^n$)

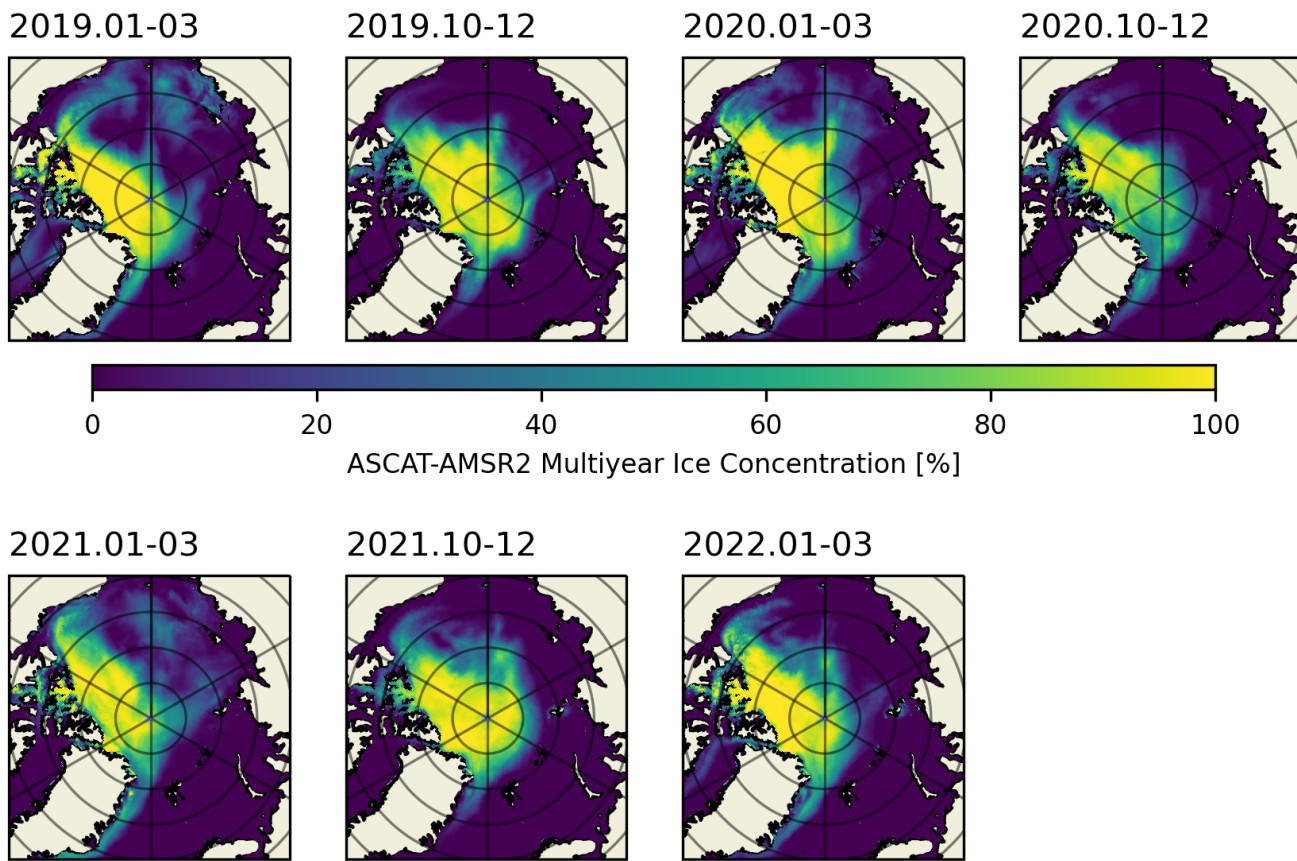

**Figure A7.** ASCAT-AMRSR2 Multiyear Ice Concentration winter three month averages for the period 2019.01-2022.03.

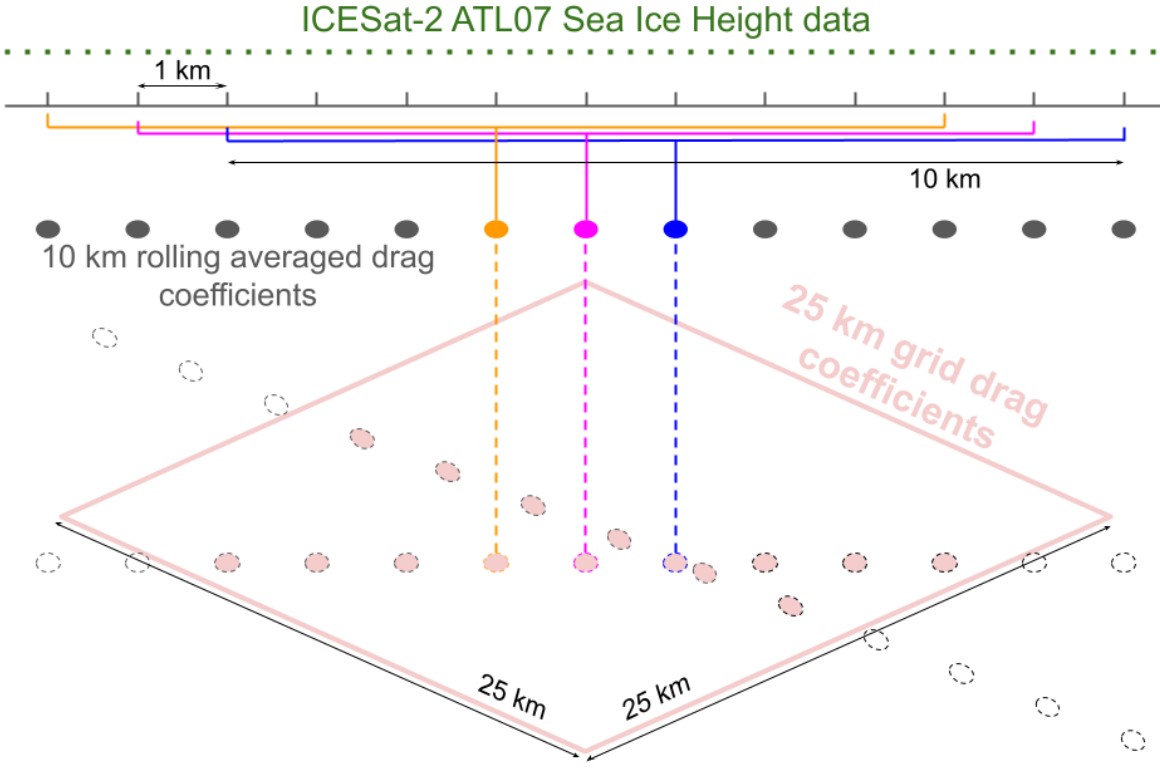

**Figure A8.** Above is a schematic showing the data processing steps. The green dotted line indicates the ATL07 sea ice height data. The grey line shows 1 km increments and the orange, pink and blue segments show typical 10 km windows over which the obstacle height and spacing are averaged (using which the drag coefficient is then calculated according to equation 3: depicted in the schematic as segments turning into dots). The resulting 10 km average drag coefficients are then gridded (e.g., the orange, pink and blue dots from the given ATL07 track are projected onto a Polar Stereographic grid, along with other values from the same track as well as those from other tracks from the same month). Finally, the values are bucket resampled to give a monthly 25 km gridded drag coefficient map (where an individual grid cell is highlighted in light red in the schametic [not to scale]).

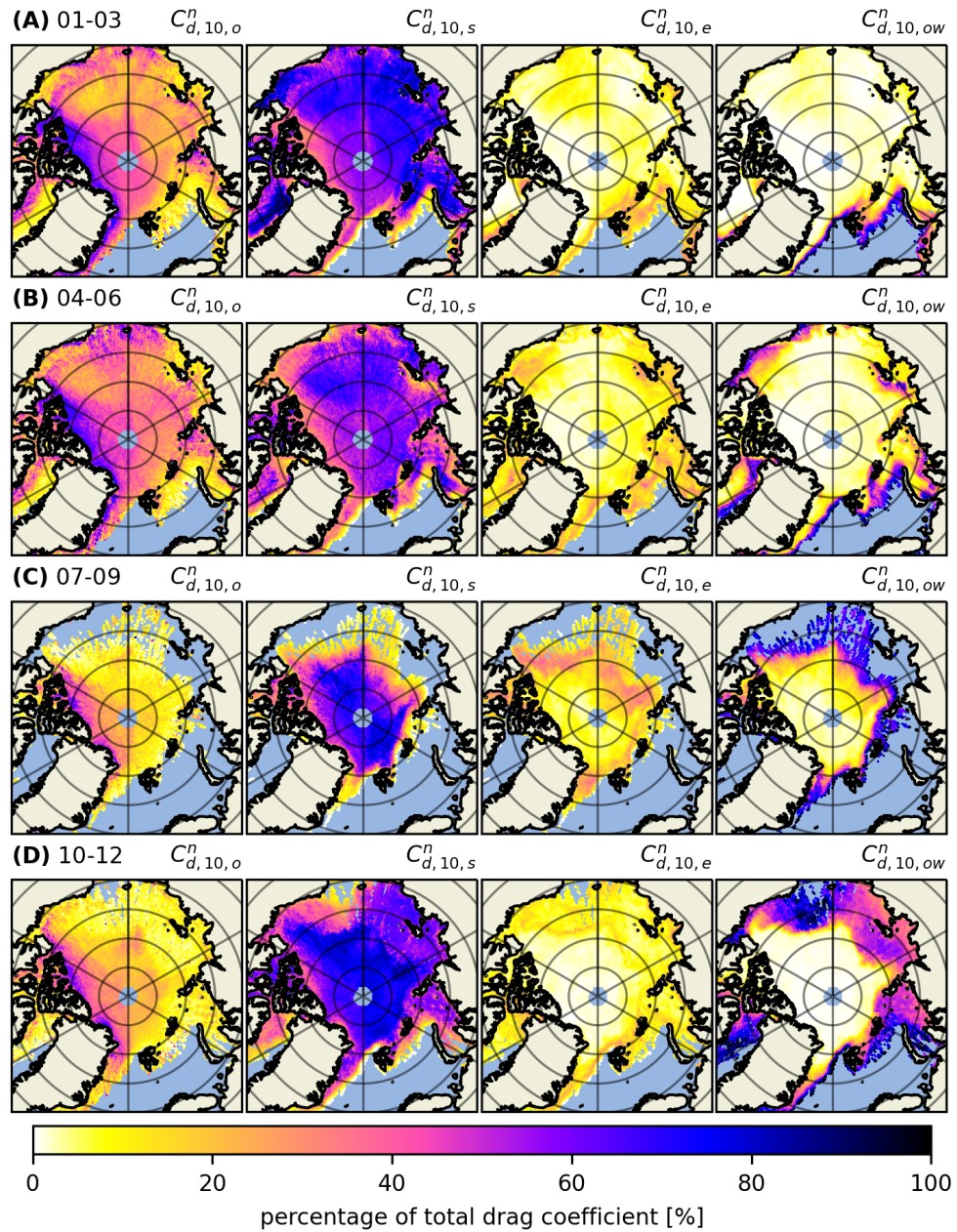

**Figure A9.** The components of the total drag coefficient given as percentages where the columns are obstacle form drag ($C_{d,10,o}^n$), sea ice skin drag ($C_{d,10,s}^n$), floe edge form drag ($C_{d,10,e}^n$) and open water skin drag ($C_{d,10,ow}^n$), respectively. These 3-monthly averages are from the year 2019 and depict the contribution of the 4 components of the total drag coefficient $C_{d,10,T}^n$ (col. 4 in Fig 5).

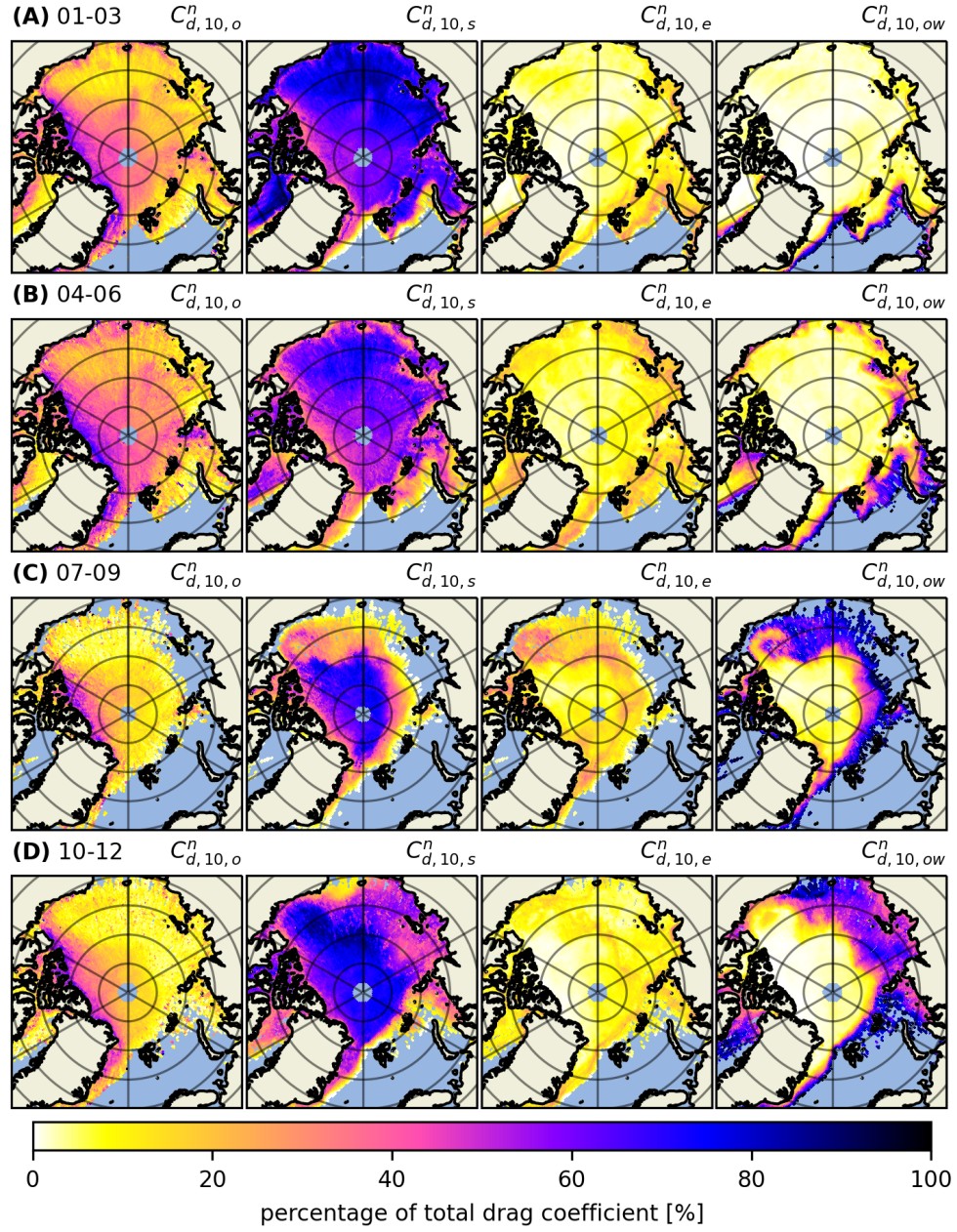

**Figure A10.** The components of the total drag coefficient given as percentages where the columns are obstacle form drag ($C_{d,10,o}^n$), sea ice skin drag ($C_{d,10,s}^n$), floe edge form drag ($C_{d,10,e}^n$) and open water skin drag ($C_{d,10,ow}^n$), respectively. These 3-monthly averages are from the year 2020 and depict the contribution of the 4 components of the total drag coefficient $C_{d,10,T}^n$ (col. 4 in Fig A5).

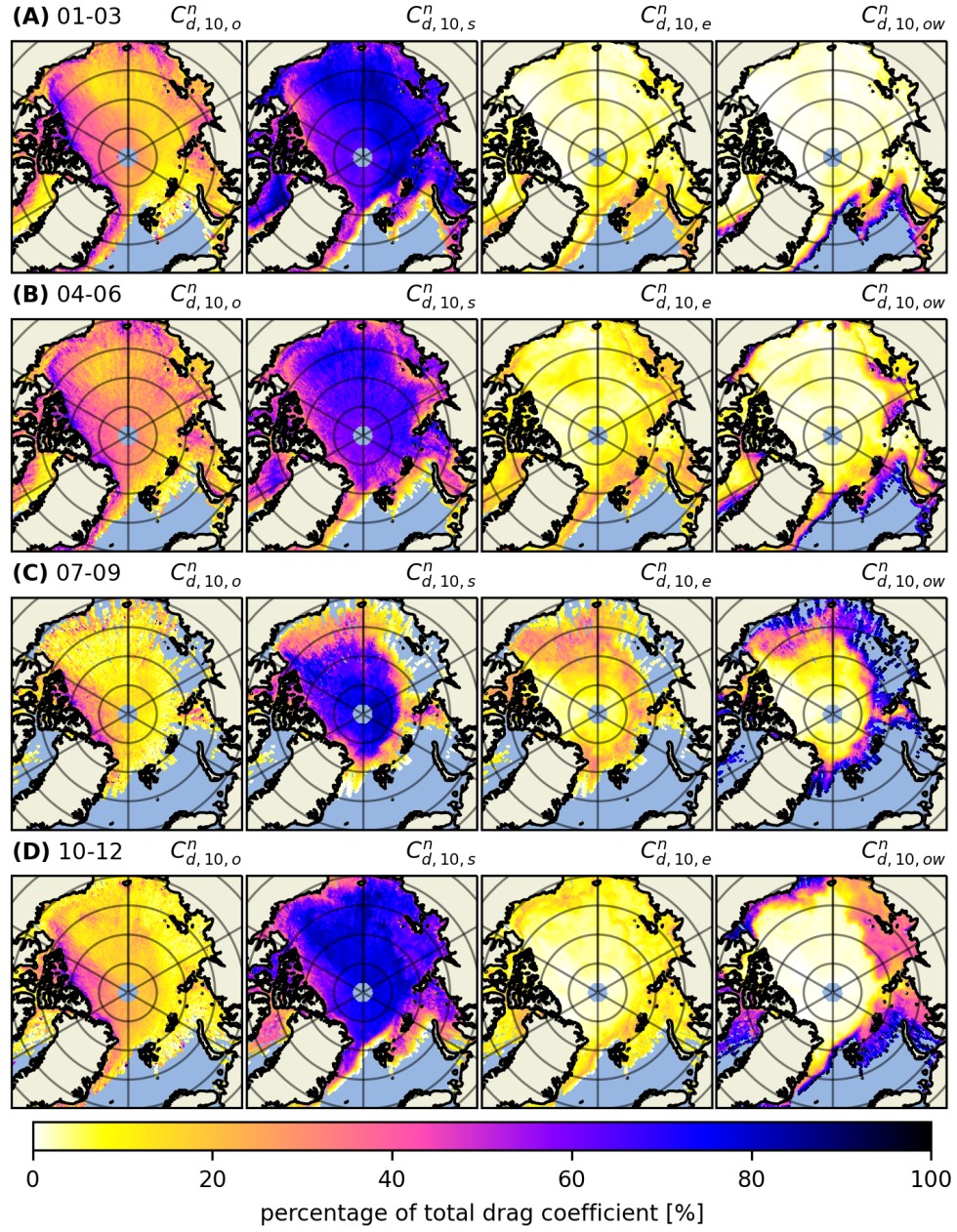

**Figure A11.** The components of the total drag coefficient given as percentages where the columns are obstacle form drag ($C_{d,10,o}^n$), sea ice skin drag ($C_{d,10,s}^n$), floe edge form drag ($C_{d,10,e}^n$) and open water skin drag ($C_{d,10,ow}^n$), respectively. These 3-monthly averages are from the year 2021 and depict the contribution of the 4 components of the total drag coefficient $C_{d,10,T}^n$ (col. 4 in Fig A6).

*Author contributions.* All authors contributed in the development of the methods used, description of relevant information and the discussion and interpretation of the results. Alexander Mchedlishvili wrote most of the paper and the main algorithm for the analysis. Christof Lüpkes provided his expertise for the parts of the paper dealing with the parameterization used and the general mathematical background behind the study. Alek Petty was conferred with for all ICESat-2-related matters, contributed to the programming behind the analysis and helped structure the paper. Michel Tsamados contributed to the programming and acted as supervisor during a brief research stay at University College London when this paper was in the works. Gunnar Spreen acted as main supervisor and helped guide the development, analysis and writing stages of this study.

*Competing interests.* The authors declare that they have no conflict of interest.

*Acknowledgements.* This research was supported by the Deutsche Forschungsgemeinschaft (DFG, German Research Foundation) within the Transregional Collaborative Research Center TRR 172 "ArctiC Amplification: Climate Relevant Atmospheric and SurfaCe Processes, and Feedback Mechanisms (AC)$^3$" (grant 268020496) and by European Union's Horizon 2020 research and innovation programme via project CRiceS (grant 101003826).

We thank NASA for providing ICESAt-2 and Operation IceBridge (OIB) data used in this study. We thank Marcus Huntemann, Robert Ricker, Marco Bagnardi, Giulia Castellani, Tom Johnson, Kyle Duncan and Sinead Farrell for their insightful discussion that helped in the preparation of this study.

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
