# Peer review of "New estimates of pan-Arctic sea ice—atmosphere neutral drag coefficients from ICESat-2 elevation data"

_EGUsphere, 2023_

## Author Comment (AC1)

**Reply to RC1**

**We thank the reviewer for their thorough reading and constructive comments, which will help to improve the manuscript. In the following, we cite the reviewers' text and add our own answers and planned text modifications in green.**

My primary concern with this study is that the derived drag coefficients are only as accurate as the Garbrecht et al. (2002) parameterization scheme used to convert from topographic measurements to drag. While Garbrecht et al. (2002) show a good comparison between parameterized and measured values of drag, the observations they use are still limited to just a few distinct, short measurement campaigns. Furthermore, the parameterization still relies on some unconstrained factors; namely the coefficients of resistance, the surface roughness length, and the effects of sheltering (different values and formulations for each of these have been suggested throughout the literature).

> **We agree, the data base is limited, but the scheme is based on more than just the Garbrecht et al. (2002) (GA02) paper. E.g., it represents a combination with the scheme of Lüpkes et al. (2012) (LU12) (as proposed by Petty et al. (2017) and as it was foreseen already by LU12). Both together reflect the present state of the art. In the revised version, we will better explain that the parametrizations of LU12 and GA02 are based on the same principal equations to determine the form drag caused by sea ice features like floe edges and ridges. The difference is just the geometry of the obstacles. In the present manuscript, we mentioned only GA02 as a validation but it was also validated based on measurement of turbulent fluxes over a single ridge (Garbrecht et al. 1999), by Vihma et al. (2003) (based on a model study) and by a PhD thesis (Ropers, 2013) based on further aircraft data of direct flux measurements. Ropers (2013) tested also different assumptions concerning the geometry of ridges and found that for closed sea ice cover a 2D geometry as used by LU12 for the marginal sea ice zone is not improving the results because of too many unknown parameters. However, since the basic idea via the calculation of the dynamical pressure is based on the same equations, one can also mention validations of the LU12 scheme as a justification of the GA02 scheme. Such a validation has been obtained by Elvidge et al. (2016) and Renfrew et al. (2019) using own aircraft flux measurements and another one based on ship measurement of fluxes by Srivastava et at al. (2022).**

Additional testing of the parameterization scheme is obviously beyond the scope of the current study and the data available. The lack of additional experimental verification of the scheme does not invalidate the results of the present study. However, the uncertainties associated with the scheme should be acknowledged by the authors, and quantified where possible. For example, how sensitive are patterns of spatial or temporal variability to the chosen formulation for the coefficient of resistance, or to the floe edge drag coefficient?

> **Sensitivity studies with different values for the coefficient of resistance (e.g., Garbrecht et al., 2002, Ropers, 2013) will be conducted. Same for the sheltering function, which will also be evaluated on the ICESat-2 data for comparison's sake.**

Would the results be similar if other schemes (e.g., one of those from Lupkes et al., 2012) were to be adopted instead? It's likely that such changes would slightly modify the drag coefficients without majorly impacting the spatial or temporal patterns of interest—but this should be tested. I recognize that previous studies that similarly convert ice topography to drag coefficients (Castellani et al., 2014; Petty et al., 2017) don't include such discussion but I nonetheless feel that it is important to acknowledge some of these unknowns.

> **We stress that the Lüpkes et al. (2012) scheme should not really be seen as an alternative because it was the idea from the very beginning of Lüpkes (here coauthor) to combine both schemes. Just for simplicity the drag coefficient for a region with 100 % sea ice cover had been prescribed to a constant value in this derivation but it was clear that in the inner Arctic especially during winter when no melt ponds are present, the topography is governed by ridges. It is described by LU12 that the assumption of constant drag coefficients over a region completely covered with sea ice is just the simplest possible approach but can be improved. An application of the LU12 scheme alone departs from the assumption of constant drag coefficients only in case of a large fraction of open water (e.g., $F_w > 10$ %). Such conditions exist only in the marginal sea ice zones or during the summer months also in the inner Arctic (mainly June – August). But already during August leads and ponds appear to be overfrozen which makes it difficult to identify such structures by remote sensing. This is the reason why in the current results, the drag coefficients caused by floe edges alone is probably underestimated. This will be better explained in the Discussion part of the revised version.**

Despite this critique, I find the subject matter and results of this study to be important, the methods and analysis to be robust, and the quality and readability of the manuscript to be good. I recommend this study for publication after revisions to address my concerns with the parameterization scheme and some further minor comments, below. I don't believe that these revisions should take a substantive effort, but am marking my recommendation as a major revision on the basis of what I feel is the importance of discussing the uncertainties associated with application of a parameterization scheme.

General comments:

- I appreciate that the study accounted for the spatial resolution of the ICESat-2 sampling by making comparisons with OIB ATM data. The authors chose to do so by developing a scaling

(eq 6) based on linear regression between the computed form drag coefficients from each data source. I am interested in the choice of regressing the computed coefficients (which depend non-linearly on each of the two topographic variables) versus regressing the topographic variables He and xe directly. Based on some of the details gleaned from Fig 1 and related text, one might expect that the mean obstacle spacing xe is most impacted by the smoothing rather than obstacle height He, so separate regressions might yield better results. Did the authors explore these different options? It is not necessary to provide a detailed analysis in the study exploring all of the different options for regressions, but the authors should acknowledge (either in §3.1 or in an appendix) if they performed these tests and either (a) overall scaling was the same; or (b) they chose to use the option with the best regression.

> **The alternate options for regression, now tested at the time of writing up this reply, will be addressed in the text and discussed.**

- The results of this study highlight the spatiotemporal variability of drag coefficients and suggest an importance of being better able to characterize obstacle statistics in numerical models. I think that there are some opportunities in the study to share some other information about the obstacle statistics that might be useful for other researchers thinking about empirically-based or simplified parameterizations. In particular, in some figures in the study the mean He and xe appear to fairly strongly negatively correlated with one another (consistent with under-ice measurements from Brenner et al, 2021; https://doi.org/10.1029/2020JC016977). I would be curious how robust such a correlation may be, or if it differs in space/time or for FYI/MYI? Furthermore, Martin (2007 Thesis; and others) suggest relationships between sail height and level ice thickness (which can be found from the modal surface level in these measurements) that might be worth sharing for these data.

> **First of all, thanks for the hint concerning the work of Brenner et al ( 2021). We will include this in the revised version. In this paper we do not aim to develop a parametrization and leave this for our future work. For this reason, we do not look into many further details. However, in the revised version we will include a discussion of the results concerning the correlation between ridge distance and ridge height. We expect that this correlation holds only for the large scale averaging, but probably not on a local basis. We will investigate this and present some graphs in the revised version considering the connection between ridge height and ridge distance on a local scale.**

> **In this connection it is important also to reflect on the limitations of ICESat-2 with regards to detecting ridges (most recently highlighted in Ricker et al. (2023)) when considering these types of correlations. The procedures which we applied, lead to a certain overestimation of ridge spacing and occasional underestimation of ridge height, referred to as "smoothing"in the paper. Corresponding scatter plots and**

**associated correlations with the pan-Arctic, MYI and FYI ice averages have been added in the appendix.**

Specific comments:

§1

- L201-22: I wholly agree with the statement that "the surface roughness of sea ice... needs to be better understood". However, the authors don't properly justify that statement here. Why is roughness important? Some of this motivation is found later, e.g., in L33-37, but I would introduce the qualitative importance of roughness before explaining it's origin.

> **We will modify the text as suggested by the reviewer, mentioning e.g. recent modeling studies by Schneider et al. (2021) as well as Yu et al. (2020) highlighting the importance of surface roughness for polar climate modelling. We will further refer to other publications who explained that surface roughness is an essential parameter in the Monin-Obukhov theory for the determination of surface fluxes and thus of the surface energy and momentum budget. We will also comment on the importance of surface roughness of operational needs of Arctic stakeholders (i.e. shipping) and local populations (i.e. safe travel on sea ice) (Dammann et al. 2018).**

- L41-45: Consider restructuring these sentences to improve the flow of the text: by starting with the definition of 'z' and its additional explanation, you disrupt the list of variables which makes it awkward to come back to the rest of them on L43. Instead shift the other items on the list (rho, U, k, theta) earlier and end with z. As a personal preference, I don't like sentences starting with a variable if it can be avoided. Rearranging the list would avoid starting a sentence with rho. Similarly, the sentence introducing the stability function, fm, can be combined with the previous one: "The drag coefficient Cd is usually written as a product of Cdn and a surface roughness dependent stability function fm"

> **The formula variable description has been modified as suggested by the reviewer.**

- L60: The statement that the parameterization has been used "successfully" can probably interpreted in multiple ways. To me, it implies that the application of the parameterization has matched observational estimates of drag; however, the cited papers don't test that. Are there any observational studies that test the parameterization aside from Garbrecht et al. (2002)?

Yes, there are such papers (see also above). As concerns comparison with independent data sets, these are the papers by Garbrecht et al. (1999) and the thesis of Ropers (2013). Furthermore, Castellani et al. (2014) show that at least the average drag coefficient, obtained from the GA02 parametrization with parameters as in our manuscript, agrees well with values given by Andreas et al. (2010) using SHEBA data. Furthermore, the approach was used by Vihma et al (2003) in a mesoscale modelling study. They showed that the application of the scheme led to a very good agreement of modelled and observed meteorological variables. Furthermore, including form drag in the marginal sea ice zones by the schemes of LU12 and Elvidge et al. (2016) resulted in an improvement of atmospheric model results described by Renfrew et al. (2019). Also Birnbaum and Lüpkes (2002) investigated the effect of floe-edge generated form drag in the marginal sea ice zone on meteorological parameters by applying a mesoscale nonhydrostatic model. The modelling can be called successful in the sense that they obtained realistic fluxes when the form drag parametrization was included and that the fraction of skin drag over form drag depends on the meteorological forcing when stability effects are accounted for. Finally, Martin et al (2016) show that the inclusion of atmospheric form drag leads to improvements in the modelling of sea ice drift. The latter work addresses only floe edge form drag but one can expect that further improvement might be possible when ridge form drag is included as well.

The term "successfully" has been omitted to avoid confusion.

§2

- L105-113: What is the effective horizontal resolution of the ATM data after processing? Does the 1m footprint result in a 1m horizontal spacing?

Yes, that is correct; that is now mentioned in the text.

- L130-134: Some type of schematic/visualization somewhere around here might be useful for understanding the data spacing referenced here and visualizing how satellite tracks fit in the 25 km gridboxes, the associated time-space variability of the data within a gridbox (i.e, colouring tracks in the schematic by time offsets or similar), and maybe the overlapping 10km segmentation.

A schematic has been produced for clarity. We thank the reviewer for pointing this out as we were not decided on whether this was necessary at first, but now see that the need for it is clear.

- L148: "...Rayleigh Criterion (explained below)" Where is the explanation for the Rayleigh criterion? I was expecting this to be defined somewhere following the numbered list, but don't see an explanation or relevant citation in the manuscript.

> **We have now added the following explanation:**
>
> **"The Rayleigh Criterion states that two maxima (obstacles) must be separated by a minimum that is less than half the value of the higher maxima for them to be classified as two separate features (e.g., hibler 1975, wadhams 1986). After omitting all elevation maxima that do not fulfil the Rayleigh Criterion, the obstacle heights and the spacing between them (both in meters) are averaged over each 10 km segment, before calculating the neutral drag coefficients at this same scale."**
>
> **We thank the reviewer for pointing it out; it must have been omitted accidentally at some point during the correction process.**

- L204-205: Mention where the cw formulation comes from (this is the one recommended by Garbrecht et al. 2002, but they also test a few other formulations). You may also be interested in seeing Zu et al., 2021 (doi.org/10.1029/2020JC016976) who use laboratory and numerical modelling in an attempt to constrain the formulation for this coefficient as applied to under-ice ridges.

> **As discussed previously, the sensitivity studies with different coefficients of resistance will now be included in the paper. We thank the reviewer for recommending reading additional literature on the topic.**

- L226-228: It might be worthwhile to provide some context for the version of the edge-drag scheme introduced here, especially given the later suggestion (L442-443) to further estimate this.

> **We have added context with the following sentence:**
>
> **"The parameterization is not fit to observations but rather derived from physical concepts and assumptions based upon drag partitioning scheme by Arya (1973, 1975)."**

- L233: In my opinion there is no need to repeat the value of Cd,s from equation 4; it's already listed on L213.

> **The repetition has been removed as suggested by the reviewer.**

§3

- L273 and Fig 2: In my opinion, it doesn't seem necessary to include the 15m and 45m filters when it is already established that the ICESat-2 data have an effective 30m spacing and the 30m filtering produces the best result. If they are kept as a part of the figure it might be helpful to include some justification about why they may be of interest to readers.

> **We have added the following justification:**
>
> **"Box sizes 15 m and 45 m are shown for comparison's sake and are meant to demonstrate how both too little and too much of this smoothing can fail to produce values comparable to that of ICESat-2."**

- Fig 2: For the no-smoothing-fit, it is hard to see the different colour contours in the heatmap (on initial look I thought it was all a single colour). For the 15m and 30m-fits it is hard to see the different colour contours in the heatmap because the 45m-fit heatmap is overlapping. In fact, I only realized that there were different colour contours because of the visibility of the core of the 45m-fit heatmap.

> **The figure has been optimized keeping the concerns of the reviewer in mind.**

- L278-280: To be clear, the drag coefficients here just the form drag coefficients, not the total drag coefficients? (Consider using the $C_{d,o}$ notation from eq. 5)

> **Yes, that is correct; the notation has been changed accordingly throughout the text.**

- Fig 3 caption: mention the equations for calculating drag coefficients in panels C, D (e.g., eq. 3 & 5 for C, and eq. 3, 5 & 6 for D).

> **The figure caption has been modified as suggested. However, importantly, equation 5 is not yet used for this figure and instead we report the drag coefficient as the sum of skin drag and form drag from obstacles (identical to the approach used for column 3 of figures 5, A3 and A4).**

- L300-305 and Fig. 4: Despite the fact that the two data sets are not perfectly co-temporal, I would initially have expected a better regressor slope than 0.5 given that the basis for much of the analysis (binning in time/space) is implicitly based on slowly-varying statistics. My initial reaction to seeing this is some suspicion that eq. 6 is not valid across all different obstacle statistics, and wondering if separate scalings of He and xe (as suggested in my general comments, above) would produce a better fit. That initial thought may prove not to

be correct, but in any case I think that the authors are fairly quick to dismiss the value of the slope. In fact, I became more hung up on the slope value than necessary since the histograms in Fig 4B and discussion in L305-310 show that the difference between the datasets is not that drastic and generally a good fit. Perhaps some reorganization here could help prevent similar hang ups?

> **We will address the mentioned issues; we will modify and rearrange the text and omit Fig. 4A as reviewer 2 has suggested.**

- Fig 4B: are the histogram bar heights probabilities? Probability densities? Counts? Label the y-axis.

> **Label has been added.**

- L333-334: Roughly what is the percent coverage of ICESat-2 data relative to total sea ice area?

> **More information on this discrepancy has been included in the text. Unfortunately, there is no study, to our knowledge, which quantifies this discrepancy. However, it is fairly clear that the total area, as a rule, will be less due to clouds and returns with ice concentrations < 15% not being processed. The more clouds over the Arctic, the higher this discrepancy will be.**

- Fig. 6A: It would be interesting to see this broken up by the contribution of all the different terms in eq. 5 (perhaps in an appendix figure), or at least similarly to columns III and IV in fig. 5 (panel A in fig. 6 corresponds to column IV in fig. 5; an additional panel could be included in fig. 6 corresponding to column III).

> **The figure mentioned here was originally discussed and considered, however, since the skin drag component is a constant and likely for the gridded data product, both the open water skin drag and the floe edge form drag components are going to exhibit the same general pattern, it seemed to be rather redundant information given figure 5 (and A3 and A4). In the data files on Pangea (cited in the reference section of this document – will be updated along with the finalized paper and re-uploaded as version 1.0), these contributions are separated – this will be mentioned in the text. If we do decide to include this type of figure, it will likely be only in the supplement.**

- Fig. 6D: Can this panel also include the total sea ice area for reference? If not reasonable to do so, I'd recommend changing the title to "Total sampled area" or something similar.

> **We have changed the title as suggested.**

- Fig. 6: Date label format is hard to read. Also, if including axis ticks for only 2 months of each year, consider using March and September to correspond to the annual sea ice maximum/minimum.

> **We have shifted the x axis to use March and September as suggested.**

- L359-368: A number of areas are mentioned here (and elsewhere in the paper) by name when describing spatial characteristics of the ice, but not everyone is familiar with these different geographic features. Including the names on a map or figure somewhere would be helpful.

> **We will add these different geographic features to figure 3 if they are clearly visible while at the same time they do not overcrowd the figure. However, we expect most researchers working on sciences associated with the Arctic to be familiar with the features mentioned. Perhaps describing their location relative to surrounding continents might also be a solution.**

- L376-377: The use of ATL03 data for similar statistical measurements should be mentioned much earlier (back in §2.1). The use of ATL07 seems well justified by the discussion in this section and it's appropriate to keep most of that discussion here, but the ability to use ATLO3 to derive roughness should be acknowledged before this section.

> **We have added the following text to figure 2.1:**
>
> **"Here we would like to mention that ATL03/L2 has been successfully used to provide more accurate information on individual ridges using an alternate processing to ATL07/L3A (see Duncan and Farrell 2022). However, for this study, we opted to use the more readily available ATL07/L3A since we are dealing with spatial averages that do not sufficiently benefit from alternate sampling routines and are in need of adjustment (discussed in section 3) regardless of the data product used."**

- §3.6: I appreciate the inclusion of this section and explicit statements of the study's significance.

- L392-393: Accounting for stability effects is (rightfully) outside the scope of the present study; however, seasonality in the surface stress in eq. 1 will depend on both seasonality in neutral drag and seasonality in the stability. It would be beneficial if the authors were to very roughly describe the impacts of stability here: specifically, would it be expected to it enhance the neutral drag coefficient seasonality or counter it?

> **We will add text addressing the stability function. But in general, over closed sea ice the near-surface stratification is more stable in winter than in summer. This means that drag coefficients will be reduced in winter relative to summer. It is more complex for fractional sea ice cover thus especially over the MIZ. There, the effect depends strongly on the sea ice fraction, temperature conditions and on wind direction (on-ice or off-ice). The reason is that form drag depends on the stratification over both sea ice and open water as shown by Lüpkes and Gryanik (2015). Thus, in this region both is possible especially during winter, namely reduction and enhancement of the drag coefficient when unstable conditions over open water have a stronger impact than the stable conditions over sea ice (see Lüpkes and Gryanik, 2015).**

- L401-402: "FYI ice peaks sometime in July-August (blue line in Fig. 6A)" This is hard to discern from figure 6A, especially as (I think) the blue line doesn't seem to even include July-August data (unless I'm misreading the date labels). The secondary peaks in ~August for each year are only in the "all ice" black lines, and are below the primary peaks. Elevated values of drag seem to exist for FYI each September.

> **Yes, that is correct and a result of the MYI masks generated from the MYI data (which are not available for some summer months). The following clarification has been included in the text as follows:**
>
> **Added clarification: "We observe that MYI ice exhibits highest drag in May (red line in Fig 6), due to an increase in the form drag due to obstacles, and FYI ice peaks sometime in July-August (according to the secondary peaks in the black line [all data] in Fig 6A and the associated presumed trajectory of the blue line [FYI data]) from a longer ice-water edge and the associated floe edge drag in summer months."**

- L410: "All other Arctic Seas (mostly FYI)" Consider including a figure showing FYI/MYI extents when available, perhaps as another column in fig. 5? (This isn't totally necessary, but would be helpful for the discussion here and in §3.3).

> **We appreciate the suggestion but do not see the need to overcrowd figure 5 with an additional column of a parameter that is not central to the study. The MYI concentration dataset as well as the associated literature is cited and the readers are invited to look at the data on the browser for spatial distributions of MYI for the time period covered.**

- L418-419: This whole study is based on the use of a form drag parameterization and frequently cites Tsamados et al., (2014) who describe the implementation of a similar parameterization in a modelling framework; however, this statement argues that further

drag parameterization is necessary. This seems contradictory and should probably be rephrased. Nonetheless, I agree that more work needs to be done in this regard; specifically, the results indicate the need for further ability to model obstacle heights/spacings and for the implementation and use of form drag parameterizations. Also see my general comment regarding additional data presentation for simplified parameterizations.

> **The Tsamados et al. (2014) implementation is not directly derived from observations but instead uses a theoretical approach. This difference will be made clearer in the text.**

§4

- L425: This is nit-picky, but I dislike the phrasing here. Saying that the study *relates* the topography to drag coefficients could be construed to mean that the two are measured independently. The study *uses* measured sea-ice topography to calculate temporal and spatial variations of atmospheric drag coefficients.

> **We agree with the concerns of the reviewer, and have modified the phrasing.**

- L443-444: Not a subject of this study, but I am curious how you plan to account for the floe sizes when considering the edge drag component. Do you intend to also determine those using ICESat-2 data (e.g, Horvat et al., 2019; https://doi.org/10.5194/tc-13-2869-2019)?

> **ATL07 includes surface classification information (lead or ice returns), and these have been shown to agree well with coincident satellite imagery from Sentinel-2 (e.g. Petty et al., 2021), opening up the possibility of deriving floe size and floe edge information directly from ICESat-2. However, the classification scheme was derived with a focus on sea surface height and freeboard derivation, and issues around misclassified dark leads (Kwok et al., 2021) and the lack of a distinct melt pond classification remain.**

> **As a first step, as discussed previously, we might include gridded freeboard from ICESat-2 to further modify the Lüpkes et al. (2012) parameterization to observer it's spatial variability.**

§A

- L457-460: I am finding this sentence hard to parse (the one beginning "As a result...").

> **Sentence has been revised as follows:**

"As a result, the regional drag coefficient estimates at higher latitudes are more representative of the time periods shown in Figs 3, 5, A3 and A5, whereas those at lower latitudes are computed with fewer height measurements (often just a few select days). In other words, rather than a temporal mean of surface topography, it is a data set that is sewn together with the best representation of the temporal mean near the pole hole."

Other grammar and typos:

- L5: Grammar error in: "though it cannot resolve as well airborne surveys"

**Corrected**

- L200: Use an in-text citation instead of a parenthetical citation for Castellani et al., 2014

**Corrected**

- L271: Suspected missing decimal in "OIB ATM occupy a wider range (~03–1.3·10^−3 )"; should probably be 0.3 instead of 03

**Yes, it was missing - now corrected**

- L300-303: Awkward sentence structure for sentence that begins: "Correlation (0.51) and slope...". Consider revising or maybe breaking up the sentence.

**Sentence restructured as follows:**

**"The correlation (0.51) and slope (0.5) of the bivariate distribution, given the two data-sets are retrieved on different days with ice drifting in between, are reasonable and positive suggesting some base similarity in the local spatial variability of surface drag."**

- Figure 6: mismatched capitalization in plot titles for panels B and C

**Corrected**

- L408: Missing figure reference

**Corrected**

**References:**

Renfrew, I. A., Elvidge, A. D., & Edwards, J. M. (2019). Atmospheric sensitivity to marginal-ice-zone drag: Local and global responses. *Quarterly Journal of the Royal Meteorological Society*, *145*(720), 1165-1179.

Srivastava, P., Brooks, I. M., Prytherch, J., Salisbury, D. J., Elvidge, A. D., Renfrew, I. A., & Yelland, M. J. (2022). Ship-based estimates of momentum transfer coefficient over sea ice and recommendations for its parameterization. *Atmospheric Chemistry and Physics*, *22*(7), 4763-4778.

Vihma, T., Hartmann, J., & Lüpkes, C. (2003). A case study of an on-ice air flow over the Arctic marginal sea-ice zone. *Boundary-layer meteorology*, *107*, 189-217.

Elvidge, A. D., Renfrew, I. A., Weiss, A. I., Brooks, I. M., Lachlan-Cope, T. A., & King, J. C. (2016). Observations of surface momentum exchange over the marginal ice zone and recommendations for its parametrisation. *Atmospheric Chemistry and Physics*, *16*(3), 1545-1563.

Garbrecht, T., Lüpkes, C., Augstein, E., & Wamser, C. (1999). Influence of a sea ice ridge on low-level airflow. *Journal of Geophysical Research: Atmospheres*, *104*(D20), 24499-24507.

Garbrecht, T., Lüpkes, C., Hartmann, J. and Wolff, M. (2002), Atmospheric drag coefficients over sea ice — validation of a parameterisation concept. Tellus A, 54: 205-219. https://doi.org/10.1034/j.1600-0870.2002.01253.x

Ropers, M. (2013). Die Auswirkung variabler Meereisrauigkeit auf die atmosphärische Grenzschicht (PhD thesis, Bremen, Universität Bremen, 2013).

Lüpkes, C., Gryanik, V. M., Hartmann, J., and Andreas, E. L. (2012), A parametrization, based on sea ice morphology, of the neutral atmospheric drag coefficients for weather prediction and climate models, *J. Geophys. Res.*, 117, D13112, doi:10.1029/2012JD017630.

Mchedlishvili, Alexander; Spreen, Gunnar; Lüpkes, Christof; Tsamados, Michel; Petty, Alek (2022): Gridded pan-Arctic total neutral atmospheric 10-m drag coefficient estimates derived from ICESat-2 ATL07 sea ice height data. PANGAEA, https://doi.org/10.1594/PANGAEA.951333

Petty, A. A., Bagnardi, M., Kurtz, N. T., Tilling, R., Fons, S., Armitage, T., et al. (2021). Assessment of ICESat-2 sea ice surface classification with Sentinel-2 imagery: Implications

for freeboard and new estimates of lead and floe geometry. Earth and Space Science, 8, e2020EA001491. https://doi.org/10.1029/2020EA001491

Kwok, R., Petty, A. A., Bagnardi, M., Kurtz, N. T., Cunningham, G. F., Ivanoff, A., and Kacimi, S.: Refining the sea surface identification approach for determining freeboards in the ICESat-2 sea ice products, The Cryosphere, 15, 821–833, https://doi.org/10.5194/tc-15-821-2021, 2021.

Schneider, T., Lüpkes, C., Dorn, W., Chechin, D., Handorf, D., Khosravi, S., ... & Rinke, A. (2022). Sensitivity to changes in the surface-layer turbulence parameterization for stable conditions in winter: A case study with a regional climate model over the Arctic. Atmospheric Science Letters, 23(1), e1066.

Yu, X., Rinke, A., Dorn, W., Spreen, G., Lüpkes, C., Sumata, H., & Gryanik, V. M. (2020). Evaluation of Arctic sea ice drift and its dependency on near-surface wind and sea ice conditions in the coupled regional climate model HIRHAM–NAOSIM. The Cryosphere, 14(5), 1727-1746.

Martin, T., Tsamados, M., Schroeder, D., & Feltham, D. L. (2016). The impact of variable sea ice roughness on changes in A rctic O cean surface stress: A model study. *Journal of Geophysical Research: Oceans*, *121*(3), 1931-1952

Ricker, R., Fons, S., Jutila, A., Hutter, N., Duncan, K., Farrell, S. L., Kurtz, N. T., and Fredensborg Hansen, R. M.: Linking scales of sea ice surface topography: evaluation of ICESat-2 measurements with coincident helicopter laser scanning during MOSAiC, EGUsphere [preprint], https://doi.org/10.5194/egusphere-2022-1122, 2022.

D. O. Dammann, H. Eicken, A. R. Mahoney, E. Saiet, F. J. Meyer and J. C. "Craig" George, "Traversing Sea Ice—Linking Surface Roughness and Ice Trafficability Through SAR Polarimetry and Interferometry," in IEEE Journal of Selected Topics in Applied Earth Observations and Remote Sensing, vol. 11, no. 2, pp. 416-433, Feb. 2018, doi: 10.1109/JSTARS.2017.2764961.

---

## Author Response (AR1)

**Reply to RC1**

**We thank the reviewer for their thorough reading and constructive comments, which has helped to improve the manuscript. In the following, we cite the reviewers' text and add our own answers and text modifications in** green.

My primary concern with this study is that the derived drag coefficients are only as accurate as the Garbrecht et al. (2002) parameterization scheme used to convert from topographic measurements to drag. While Garbrecht et al. (2002) show a good comparison between parameterized and measured values of drag, the observations they use are still limited to just a few distinct, short measurement campaigns. Furthermore, the parameterization still relies on some unconstrained factors; namely the coefficients of resistance, the surface roughness length, and the effects of sheltering (different values and formulations for each of these have been suggested throughout the literature).

> **We agree, the database is limited, but more has been done towards validation than in the work of Garbrecht et al. (2002) (GA02). To understand this, it is first important to see that the parameterization is a combination of the schemes of GA02 and of Lüpkes et al. (2012) (LU12) (as proposed by Petty et al., 2017). Both schemes together reflect the present state of the art. Since the parametrizations of LU12 and GA02 are based on the same principal equations and differences refer only to the assumed geometry of obstacles, the validation work of the LU12 scheme can also be taken into account here to some extent. Moreover, the GA02 scheme was not only validated by GA02 but also already by Garbrecht et al (1999) based on measurement of turbulent fluxes over a single ridge, by Vihma et al. (2003) (based on a model study) and by a PhD thesis (Ropers, 2013) based on airborne turbulence measurements. Ropers (2013) tested furthermore different assumptions concerning the geometry of ridges and found that for closed sea ice cover a 2D geometry for ridges as used by LU12 for sea ice floes is not improving the results because of too many unknown parameters. A validation of the LU12 scheme has been obtained independent from the LU12 own validation by Elvidge et al. (2016) and by Renfrew et al. (2019) and by Srivastava et al. (2022) who used own aircraft and ship flux measurements respectively of fluxes.**
>
> **All this is included now in the new subsection 3.6**

Additional testing of the parameterization scheme is obviously beyond the scope of the current study and the data available. The lack of additional experimental verification of the scheme does not invalidate the results of the present study. However, the uncertainties associated with the scheme should be acknowledged by the authors, and quantified where possible. For example, how sensitive are patterns of spatial or temporal variability to the chosen formulation for the coefficient of resistance, or to the floe edge drag coefficient?

> **To address the concerns of the reviewer we have conducted a sensitivity study with different values for the coefficient of resistance (e.g., Garbrecht et al., 2002,**

**Ropers, 2013). The results of this study are now shown in figure A4. Sensitivity studies with the sheltering function have been evaluated and its effects are briefly reported in the new "Discussion of Methods" section (3.6). In this section, there is also added discussion on the two parameterizations used (LU12 and GA02) exploring their uncertainties and the advantages of the combined usage of both.**

Would the results be similar if other schemes (e.g., one of those from Lupkes et al., 2012) were to be adopted instead? It's likely that such changes would slightly modify the drag coefficients without majorly impacting the spatial or temporal patterns of interest—but this should be tested. I recognize that previous studies that similarly convert ice topography to drag coefficients (Castellani et al., 2014; Petty et al., 2017) don't include such discussion but I nonetheless feel that it is important to acknowledge some of these unknowns.

**We stress that the Lüpkes et al. (2012) (LU12) scheme should not really be seen as an alternative because it was the idea from the very beginning of Lüpkes (here coauthor) to combine both schemes. Just for simplicity the drag coefficient for a region with 100 % sea ice cover had been prescribed to a constant value in this derivation but it was clear that in the inner Arctic especially during winter when no melt ponds are present, the topography is governed by ridges. It is described by LU12 that the assumption of constant drag coefficients over a region completely covered with sea ice is just the simplest possible approach but can be improved. An application of the LU12 scheme alone departs from the assumption of constant drag coefficients only in case of a large fraction of open water (e.g., $F_w$ > 10 %). Such conditions exist only in the marginal sea ice zones or during the summer months also in the inner Arctic (mainly June – August). But already during August leads and ponds appear to be overfrozen which makes it difficult to identify such structures by remote sensing. This is the reason why in the current results, the drag coefficients caused by floe edges alone are probably underestimated. We stress the idea of a combined use of the schemes GA02 and LU12 in the new subsection 3.6.**

Despite this critique, I find the subject matter and results of this study to be important, the methods and analysis to be robust, and the quality and readability of the manuscript to be good. I recommend this study for publication after revisions to address my concerns with the parameterization scheme and some further minor comments, below. I don't believe that these revisions should take a substantive effort, but am marking my recommendation as a major revision on the basis of what I feel is the importance of discussing the uncertainties associated with application of a parameterization scheme.

**Thank you for this positive conclusion. We significantly extended the manuscript regarding the parametrization scheme and hope the open questions are answered now.**

General comments:

 - I appreciate that the study accounted for the spatial resolution of the ICESat-2 sampling by making comparisons with OIB ATM data. The authors chose to do so by developing a scaling (eq 6) based on linear regression between the computed form drag coefficients from each data source. I am interested in the choice of regressing the computed coefficients (which depend non-linearly on each of the two topographic variables) versus regressing the topographic variables $H_e$ and $x_e$ directly. Based on some of the details gleaned from Fig 1 and related text, one might expect that the mean obstacle spacing $x_e$ is most impacted by the smoothing rather than obstacle height $H_e$, so separate regressions might yield better results. Did the authors explore these different options? It is not necessary to provide a detailed analysis in the study exploring all of the different options for regressions, but the authors should acknowledge (either in §3.1 or in an appendix) if they performed these tests and either (a) overall scaling was the same; or (b) they chose to use the option with the best regression.

**The alternate options for regression are described as follows in the text: "Here we focused on comparing the average drag coefficients from satellite and airborne instruments rather than the component parameters: obstacle height and obstacle spacing. The reason for this approach is because that is where the best regression was found. Regressing obstacle heights shows decent agreement but very little difference when evaluating the moving average filter test (as shown in Fig. 1 of the manuscript). Evaluating the different average filter box sizes on the OIB obstacle heights shows very small differences (see Fig. I below). The reason for this is because while the smoothing introduced in ATL07 effectively retrieves the tall narrow ridges as smaller than they really are, this also pushes a lot of small ridges below the cutoff, reducing the sample size.**

[Figure]

I. Heat maps of 12.5 km grid resampled 10-km average ICESat-2 ATL07 avg. ridge heights plotted against those computed from OIB ATM drag coefficients from 4, 8, 19 and 22 April 2019; resampled and calculated in the same manner (in blue) as well as the OIB ATM data smoothed by a moving average filter with a window sizes of 15 m (in orange), 30 m (in green) and 45 m (in pink). The lines represent Huber fits with colour coding matching that of the bivariate heat maps; except for the dashed black

line which represents the identity line.

This reduction results in similar averaged values between the smoothed and high-resolution data-sets as can be seen in Figs. 2A and 2B, where the average obstacle height H_e is the same. The only exception are the features that are not detected at all (Ricker et al., 2023), which force the regression to be steeper than expected.

Obstacle spacing on the other hand (see Fig. II below), is where the smoothing really gets in the way of extracting any meaningful relationship. As can be seen in Figs. 1A and 1B of the manuscript, the smoothing reduces sample size which is directly proportional to obstacle spacing, as less obstacles translates to a higher average spacing between them.

[Figure]

II. Heat maps of 12.5 km grid resampled 10-km average ICESat-2 ATL07 avg. ridge spacing plotted against those computed from OIB ATM drag coefficients from 4, 8, 19 and 22 April 2019; resampled and calculated in the same manner (in blue) as well as the OIB ATM data smoothed by a moving average filter with a window sizes of 15 m (in orange), 30 m (in green) and 45 m (in pink). The lines represent Huber fits with colour coding matching that of the bivariate heat maps; except for the dashed black line which represents the identity line.

By evaluating equation 3 with the input parameters, we see how drag coefficients derived from ICESat-2 ATL07 topography can be simulated with the higher resolution OIB ATM data (Fig. 3) , as can be seen also in Fig. 2. Although this effect is partly visible in the obstacle spacing heatmap scatter plots shown in the figure above (II), the regressions produced are not meaningful since the data spread is very large. Thus, we again propose using the previous approach and regressing at the drag coefficient level so as to extract a meaningful regression.

- The results of this study highlight the spatiotemporal variability of drag coefficients and suggest an importance of being better able to characterize obstacle statistics in numerical models. I think that there are some opportunities in the study to share some other information about the obstacle statistics that might be useful for other researchers thinking about empirically-based or simplified parameterizations. In particular, in some figures in the study the mean He and xe appear to fairly strongly negatively correlate with one another (consistent with under-ice measurements from Brenner et al, 2021; https://doi.org/10.1029/2020JC016977). I would be curious how robust such a correlation may be, or if it differs in space/time or for FYI/MYI? Furthermore, Martin (2007 Thesis; and others) suggest relationships between sail height and level ice thickness (which can be found from the modal surface level in these measurements) that might be worth sharing for these data.

> **First of all, thanks for the hint concerning the work of Brenner et al (2021), it is now mentioned in the revised version. In this paper we do not aim to develop a parametrization from the obstacle statistics and leave this for our future work.**
>
> **In spite of this limitation, the questions brought up by the reviewer have inspired us to share some of the correlations found from the obstacle statistics produced for this study (Fig. A3). For the pan-Arctic, MYI and FYI averages of obstacle height and spacing averages (presented in Fig. 4), we have indeed found a negative correlation of -0.87 (mentioned in Fig. A3 of the manuscript), corroborating the findings of Brenner et al ( 2021).**
>
> **As expected, the smoothing caused by ICESat-2 ATL07 typically overestimates ridge spacing Arctic-wide, however, we expect the correlations we found to be valid for large spatial scales.**

Specific comments:

§1

- L201-22: I wholly agree with the statement that "the surface roughness of sea ice... needs to be better understood". However, the authors don't properly justify that statement here. Why is roughness important? Some of this motivation is found later, e.g., in L33-37, but I would introduce the qualitative importance of roughness before explaining it's origin.

> **The following text on the subject of importance of sea ice roughness has been added:**
>
> **"Surface roughness can be related to the neutral drag coefficient by applying the Monin-Obukhov theory. Since the roughness length for momentum and the scalar**

**roughness length for heat and moisture occur also in the non-neutral transfer coefficients, surface roughness directly impacts not only momentum transport but also the transfer of heat and moisture between the atmosphere and the underlying surface. Rougher surfaces can create more turbulence and enhance mixing, thereby influencing the stability of the atmospheric boundary layer (e.g. Garratt, 1992; Schneider et al., 2022; Lüpkes and Gryanik, 2015). Due to its impact on momentum and heat transport over and below the sea ice layer, surface roughness is a fundamental parameter influencing the distribution of sea ice (e.g., Yu et al., 2020; Brenner et al., 2021)."**

**And later on:**

**"Rougher sea ice is generally found in the thick, multiyear ice regions north of the Arctic Archipelago and Greenland. Landfast rough ice in these areas is an important factor for determining transportation routes for local residents and industry (Dammann et al., 2018)."**

- L41-45: Consider restructuring these sentences to improve the flow of the text: by starting with the definition of 'z' and its additional explanation, you disrupt the list of variables which makes it awkward to come back to the rest of them on L43. Instead shift the other items on the list (rho, U, k, theta) earlier and end with z. As a personal preference, I don't like sentences starting with a variable if it can be avoided. Rearranging the list would avoid starting a sentence with rho. Similarly, the sentence introducing the stability function, fm, can be combined with the previous one: "The drag coefficient Cd is usually written as a product of Cdn and a surface roughness dependent stability function fm"

> **The formula variable description has been modified as suggested by the reviewer.**

- L60: The statement that the parameterization has been used "successfully" can probably interpreted in multiple ways. To me, it implies that the application of the parameterization has matched observational estimates of drag; however, the cited papers don't test that. Are there any observational studies that test the parameterization aside from Garbrecht et al. (2002)?

> **Yes, there are such papers (see also above). As concerns comparison with independent data sets, these are the papers by Garbrecht et al. (1999) and the thesis of Ropers (2013). Furthermore, Castellani et al. (2014) show that at least the average drag coefficient, obtained from the GA02 parametrization with parameters as in our manuscript, agrees well with values given by Andreas et al. (2010) using SHEBA data. Furthermore, the GA02 parameterization was used by Vihma et al (2003) in a mesoscale modelling study. They showed that the application of the scheme led to a very good agreement of modelled and observed meteorological variables. Furthermore, work is done to validate the principal approach describing form drag on sea ice. So, the LU12 for form drag approach was confirmed by**

Elvidge et al. (201for form drag in the marginal sea ice zones. Renfrew et al. (2019) approach resulted in an improvement of atmospheric model results described by Renfrew et al. (2019). Also Birnbaum and Lüpkes (2002) investigated the effect of floe-edge generated form drag in the marginal sea ice zone on meteorological parameters by applying a mesoscale nonhydrostatic model. The modelling can be called successful in the sense that they obtained realistic fluxes when the form drag parameterization was included and that the fraction of skin drag over form drag depends on the meteorological forcing when stability effects are accounted for. Finally, Martin et al (2016) show that the inclusion of atmospheric form drag leads to improvements in the modelling of sea ice drift. The latter work addresses only floe edge form drag but one can expect that further improvement might be possible when ridge form drag is included as well.

The term "successfully" has been omitted to avoid confusion.

§2

- L105-113: What is the effective horizontal resolution of the ATM data after processing? Does the 1m footprint result in a 1m horizontal spacing?

Yes, that is correct; that is now mentioned in the text.

- L130-134: Some type of schematic/visualization somewhere around here might be useful for understanding the data spacing referenced here and visualizing how satellite tracks fit in the 25 km gridboxes, the associated time-space variability of the data within a gridbox (i.e, colouring tracks in the schematic by time offsets or similar), and maybe the overlapping 10km segmentation.

A schematic has been produced for clarity and added to the paper with supporting text (Fig. A8). We thank the reviewer for pointing this out as we were not decided on whether this was necessary at first, but now see that the need for it is clear.

- L148: "...Rayleigh Criterion (explained below)" Where is the explanation for the Rayleigh criterion? I was expecting this to be defined somewhere following the numbered list, but don't see an explanation or relevant citation in the manuscript.

We have now added the following explanation:

"The Rayleigh Criterion states that two maxima (obstacles) must be separated by a minimum that is less than half the value of the higher maxima for them to be classified as two separate features (e.g.,Hibler 1975, Wadhams 1986). After omitting all elevation maxima that do not fulfill the Rayleigh Criterion, the obstacle heights and the spacing between them (both in meters) are averaged over each 10 km segment, before calculating the neutral drag coefficients at this same scale."

**We thank the reviewer for pointing it out; it must have been omitted accidentally at some point during the correction process.**

- L204-205: Mention where the cw formulation comes from (this is the one recommended by Garbrecht et al. 2002, but they also test a few other formulations). You may also be interested in seeing Zu et al., 2021 (doi.org/10.1029/2020JC016976) who use laboratory and numerical modelling in an attempt to constrain the formulation for this coefficient as applied to under-ice ridges.

**The sensitivity studies with different coefficients of resistance are now included in the paper, in the appendix. We thank the reviewer for recommending reading additional literature on the topic.**

- L226-228: It might be worthwhile to provide some context for the version of the edge-drag scheme introduced here, especially given the later suggestion (L442-443) to further estimate this.

**We have added context with the following sentence:**

**"The parameterization does not just represent a simple fit to observations but was rather derived from physical concepts and assumptions based upon drag partitioning scheme by Arya (1973, 1975)."**

- L233: In my opinion there is no need to repeat the value of Cd,s from equation 4; it's already listed on L213.

**The repetition has been removed as suggested by the reviewer.**

- L273 and Fig 2: In my opinion, it doesn't seem necessary to include the 15m and 45m filters when it is already established that the ICESat-2 data have an effective 30m spacing and the 30m filtering produces the best result. If they are kept as a part of the figure it might be helpful to include some justification about why they may be of interest to readers.

**We have added the following justification:**

**"Box sizes 15 m and 45 m are shown for comparison's sake and are meant to demonstrate how both too little and too much of this smoothing can fail to produce values comparable to that of ICESat-2."**

- Fig 2: For the no-smoothing-fit, it is hard to see the different colour contours in the heatmap (on initial look I thought it was all a single colour). For the 15m and 30m-fits it is hard to see the different colour contours in the heatmap because the 45m-fit heatmap is

overlapping. In fact, I only realized that there were different colour contours because of the visibility of the core of the 45m-fit heatmap.

> **The figure has been optimised keeping the concerns of the reviewer in mind. Specifically, the heatmap contours no longer have a common norm thereby better showing the variability within each individual heatmap.**

- L278-280: To be clear, the drag coefficients here just the form drag coefficients, not the total drag coefficients? (Consider using the Cd,o notation from eq. 5)

> **Yes, that is correct; the notation has been changed accordingly throughout the text.**

- Fig 3 caption: mention the equations for calculating drag coefficients in panels C, D (e.g., eq. 3 & 5 for C, and eq. 3, 5 & 6 for D).

> **The figure caption has been modified as suggested. However, importantly, equation 5 is not yet used for this figure and instead we report the drag coefficient as the sum of skin drag and form drag from obstacles (identical to the approach used for column 3 of figures 5, A3 and A4).**

§3

- L300-305 and Fig. 4: Despite the fact that the two data sets are not perfectly co-temporal, I would initially have expected a better regressor slope than 0.5 given that the basis for much of the analysis (binning in time/space) is implicitly based on slowly-varying statistics. My initial reaction to seeing this is some suspicion that eq. 6 is not valid across all different obstacle statistics, and wondering if separate scalings of He and xe (as suggested in my general comments, above) would produce a better fit. That initial thought may prove not to be correct, but in any case I think that the authors are fairly quick to dismiss the value of the slope. In fact, I became more hung up on the slope value than necessary since the histograms in Fig 4B and discussion in L305-310 show that the difference between the datasets is not that drastic and generally a good fit. Perhaps some reorganization here could help prevent similar hang ups?

> **The text has been reorganised and Fig. 4A has been omitted as reviewer 2 had suggested. With regards to the regression at the level of $H_e$ and $X_e$, please see our reply to your general comments.**

- Fig 4B: are the histogram bar heights probabilities? Probability densities? Counts? Label the y-axis.

> **A label has been added.**

- L333-334: Roughly what is the percent coverage of ICESat-2 data relative to total sea ice area?

**More information on the sampling strategy and cloud masking has been included in the text. Unfortunately, there is no study, to our knowledge, which quantifies this effect. However, it is fairly clear that the total area, as a rule, will be less due to clouds and returns with ice concentrations < 15% not being processed. The more clouds over the Arctic, the higher this reduction in area will be.**

- Fig. 6A: It would be interesting to see this broken up by the contribution of all the different terms in eq. 5 (perhaps in an appendix figure), or at least similarly to columns III and IV in fig. 5 (panel A in fig. 6 corresponds to column IV in fig. 5; an additional panel could be included in fig. 6 corresponding to column III).

**Thank you for the idea. Figures for the years 2019, 2020 and 2021 that show this component breakdown as percentages of the total drag coefficient (already presented in Figs. 5, A5 and A6) are now in the appendix (Figs. A9-A11).**

- Fig. 6D: Can this panel also include the total sea ice area for reference? If not reasonable to do so, I'd recommend changing the title to "Total sampled area" or something similar.

**We have changed the title as suggested.**

- Fig. 6: Date label format is hard to read. Also, if including axis ticks for only 2 months of each year, consider using March and September to correspond to the annual sea ice maximum/minimum.

**We have shifted the x axis to use March and September as suggested.**

- L359-368: A number of areas are mentioned here (and elsewhere in the paper) by name when describing spatial characteristics of the ice, but not everyone is familiar with these different geographic features. Including the names on a map or figure somewhere would be helpful.

**Adding place names to maps that display pan-Arctic data proved to be overcrowding. As a result, we have included additional information in brackets next to some place names which should hopefully make everything a bit clearer.**

- L376-377: The use of ATL03 data for similar statistical measurements should be mentioned much earlier (back in §2.1). The use of ATL07 seems well justified by the discussion in this

section and it's appropriate to keep most of that discussion here, but the ability to use ATLO3 to derive roughness should be acknowledged before this section.

> **We have added the following text to figure 2.1:**

> **"Here we would like to mention that ATL03/L2 has been successfully used to provide more accurate information on individual ridges using an alternate processing to ATL07/L3A (see Duncan and Farrell 2022). However, for this study, we opted to use the more readily available ATL07/L3A since we are dealing with spatial averages that do not sufficiently benefit from alternate sampling routines and are in need of adjustment (discussed in section 3) regardless of the data product used."**

- §3.6: I appreciate the inclusion of this section and explicit statements of the study's significance.

- L392-393: Accounting for stability effects is (rightfully) outside the scope of the present study; however, seasonality in the surface stress in eq. 1 will depend on both seasonality in neutral drag and seasonality in the stability. It would be beneficial if the authors were to very roughly describe the impacts of stability here: specifically, would it be expected to it enhance the neutral drag coefficient seasonality or counter it?

> **We have added the following text to address the points brought up by the reviewer:**

> **"As mentioned previously in section 1, the drag coefficient $C_d$ also depends on the surface roughness dependent stability function $f_m$, for which numerous versions exist (see e.g. Gryanik and Lüpkes (2018, 2022). For this study we have limited our research to assessing the neutral drag coefficients $C_{nd}$. In case of stable stratification $C_d$ becomes smaller than $C_{nd}$, whereas unstable stratification with more turbulence causes $C_d$ to be greater than $C_{nd}$ (Lüpkes and Gryanik, 2015). The local near-surface stratification is heavily impacted by open-water that facilitates upward heat fluxes (Andreas and Cash, 1999; Lüpkes and Gryanik, 2015) and as a result varies between the more ice-covered inner Arctic and the MIZ where open water is more common. Thus, it is in summer, where more open water is present across the Arctic ice cap, that our estimates of the neutral drag coefficients $C_{nd}$ are likely below $C_d$ . Conversely, over regions with large sea ice cover the stratification is expected to be more stable in winter during polar nights (Lüpkes and Gryanik, 2015), which will act to offset the impact of higher form drag. Suggesting our estimates of $C_{nd}$ for winter are more representative of $C_d$."**

- L401-402: "FYI ice peaks sometime in July-August (blue line in Fig. 6A)" This is hard to discern from figure 6A, especially as (I think) the blue line doesn't seem to even include July-August data (unless I'm misreading the date labels). The secondary peaks in ~August for each year are only in the "all ice" black lines, and are below the primary peaks. Elevated values of drag seem to exist for FYI each September.

> **Yes, that is correct and a result of the MYI masks generated from the MYI data (which are not available for some summer months). The following clarification has been included in the text as follows:**

> **Added clarification: "We observe that MYI ice exhibits highest drag in May (red line in Fig 6), due to an increase in the form drag due to obstacles, and FYI ice peaks sometime in July-August (according to the secondary peaks in the black line [all data] in Fig 6A and the associated presumed trajectory of the blue line [FYI data]) from a longer ice-water edge and the associated floe edge drag in summer months."**

- L410: "All other Arctic Seas (mostly FYI)" Consider including a figure showing FYI/MYI extents when available, perhaps as another column in fig. 5? (This isn't totally necessary, but would be helpful for the discussion here and in §3.3).

> **We appreciate the suggestion and have created a separate appendix figure of the available three monthly means of MYI concentration to compliment Figs. 5, A5 and A6. We avoided doing this on top of the former figures so as not to overcrowd them.**

- L418-419: This whole study is based on the use of a form drag parameterization and frequently cites Tsamados et al., (2014) who describe the implementation of a similar parameterization in a modelling framework; however, this statement argues that further drag parameterization is necessary. This seems contradictory and should probably be rephrased. Nonetheless, I agree that more work needs to be done in this regard; specifically, the results indicate the need for further ability to model obstacle heights/spacings and for the implementation and use of form drag parameterizations. Also see my general comment regarding additional data presentation for simplified parameterizations.

> **The Tsamados et al. (2014) implementation is not directly derived from observations but instead uses a theoretical approach. This is also now reiterated in the text. It is our hope and belief that more studies using theoretical parameterizations with high-resolution observations will provide needed baselines to help develop and potentially improve existing parameterization schemes.**

§4

- L425: This is nit-picky, but I dislike the phrasing here. Saying that the study *relates* the topography to drag coefficients could be construed to mean that the two are measured independently. The study *uses* measured sea-ice topography to calculate temporal and spatial variations of atmospheric drag coefficients.

**We agree with the concerns of the reviewer, and have modified the phrasing.**

- L443-444: Not a subject of this study, but I am curious how you plan to account for the floe sizes when considering the edge drag component. Do you intend to also determine those using ICESat-2 data (e.g, Horvat et al., 2019; https://doi.org/10.5194/tc-13-2869-2019)?

**ATL07 includes surface classification information (lead or ice returns), and these have been shown to agree well with coincident satellite imagery from Sentinel-2 (e.g. Petty et al., 2021), opening up the possibility of deriving floe size and floe edge information directly from ICESat-2. However, the classification scheme was derived with a focus on sea surface height and freeboard derivation, and issues around misclassified dark leads (Kwok et al., 2021) and the lack of a distinct melt pond classification remain.**

**As a first step, as discussed previously, we have tried including a gridded freeboard from ICESat-2 to further modify the Lüpkes et al. (2012) parameterization to observe its spatial variability. We touched on this experiment in the updated text as follows:**

**"Here, we tested a different hierarchy level of the Lüpkes et al. (2012) scheme than that which is used for this study (level 4). It is their level 2 parameterization which allows for specifying the measured grid-cell averaged freeboard. Instead of the constant value 0.41 m that is implicitly used in the Lüpkes et al. (2012) version used in the previous sections, we considered the data from ATLAS/ICESat-2 L3B Daily and Monthly Gridded Sea Ice Freeboard, Version 3 (ATL20) thereby implementing freeboard from satellite remote sensing measurements. Because of the smoothing imposed by sampling the results did not show any significant improvement over using constant freeboard hf= 0.41m as recommended in Lüpkes et al. (2012) for the simpler level. Ideally, all components of floe edge form drag coefficients should be taken from remote sensing to better monitor the changing Arctic system, but especially with regards to floe edge sizes, ICESat-2 cannot reliably determine this parameter Arctic-wide. Though it is beyond the scope of this study, we encourage future work in this direction with a multi-satellite approach that might remedy the limitations of each individual instrument."**

§A

- L457-460: I am finding this sentence hard to parse (the one beginning "As a result...").

**Sentence has been revised as follows:**

**"As a result, the regional drag coefficient estimates at higher latitudes are more representative of the time periods shown in Figs 3, 5, A3 and A5, whereas those at lower latitudes are computed with fewer height measurements (often just a few select days). In other words, rather than a temporal mean of surface topography, it is a data set that is sewn together with the best representation of the temporal mean near the pole hole."**

Other grammar and typos:

- L5: Grammar error in: "though it cannot resolve as well airborne surveys"

**Corrected**

- L200: Use an in-text citation instead of a parenthetical citation for Castellani et al., 2014

**Corrected**

- L271: Suspected missing decimal in "OIB ATM occupy a wider range (~03−1.3·10^−3 )"; should probably be 0.3 instead of 03

**Yes, it was missing - now corrected**

- L300-303: Awkward sentence structure for sentence that begins: "Correlation (0.51) and slope...". Consider revising or maybe breaking up the sentence.

**Sentence restructured as follows:**

**"The correlation (0.51) and slope (0.5) of the bivariate distribution, given the two data-sets are retrieved on different days with ice drifting in between, are reasonable and positive suggesting some base similarity in the local spatial variability of surface drag."**

- Figure 6: mismatched capitalization in plot titles for panels B and C

**Corrected**

- L408: Missing figure reference

**Corrected**

**References:**

Renfrew, I. A., Elvidge, A. D., & Edwards, J. M. (2019). Atmospheric sensitivity to marginal-ice-zone drag: Local and global responses. *Quarterly Journal of the Royal Meteorological Society*, *145*(720), 1165-1179.

Srivastava, P., Brooks, I. M., Prytherch, J., Salisbury, D. J., Elvidge, A. D., Renfrew, I. A., & Yelland, M. J. (2022). Ship-based estimates of momentum transfer coefficient over sea ice and recommendations for its parameterization. *Atmospheric Chemistry and Physics*, *22*(7), 4763-4778.

Vihma, T., Hartmann, J., & Lüpkes, C. (2003). A case study of an on-ice air flow over the Arctic marginal sea-ice zone. *Boundary-layer meteorology*, *107*, 189-217.

Elvidge, A. D., Renfrew, I. A., Weiss, A. I., Brooks, I. M., Lachlan-Cope, T. A., & King, J. C. (2016). Observations of surface momentum exchange over the marginal ice zone and recommendations for its parametrisation. *Atmospheric Chemistry and Physics*, *16*(3), 1545-1563.

Garbrecht, T., Lüpkes, C., Augstein, E., & Wamser, C. (1999). Influence of a sea ice ridge on low-level airflow. *Journal of Geophysical Research: Atmospheres*, *104*(D20), 24499-24507.

Garbrecht, T., Lüpkes, C., Hartmann, J. and Wolff, M. (2002), Atmospheric drag coefficients over sea ice — validation of a parameterisation concept. Tellus A, 54: 205-219. https://doi.org/10.1034/j.1600-0870.2002.01253.x

Ropers, M. (2013). Die Auswirkung variabler Meereisrauigkeit auf die atmospha̋rische Grenzschicht (PhD thesis, Bremen, Universität Bremen, 2013).

Lüpkes, C., Gryanik, V. M., Hartmann, J., and Andreas, E. L. (2012), A parametrization, based on sea ice morphology, of the neutral atmospheric drag coefficients for weather prediction and climate models, *J. Geophys. Res.*, 117, D13112, doi:10.1029/2012JD017630.

Mchedlishvili, Alexander; Spreen, Gunnar; Lüpkes, Christof; Tsamados, Michel; Petty, Alek (2022): Gridded pan-Arctic total neutral atmospheric 10-m drag coefficient estimates derived from ICESat-2 ATL07 sea ice height data. PANGAEA, https://doi.org/10.1594/PANGAEA.951333

Petty, A. A., Bagnardi, M., Kurtz, N. T., Tilling, R., Fons, S., Armitage, T., et al. (2021). Assessment of ICESat-2 sea ice surface classification with Sentinel-2 imagery: Implications for freeboard and new estimates of lead and floe geometry. Earth and Space Science, 8, e2020EA001491. https://doi.org/10.1029/2020EA001491

Kwok, R., Petty, A. A., Bagnardi, M., Kurtz, N. T., Cunningham, G. F., Ivanoff, A., and Kacimi, S.: Refining the sea surface identification approach for determining freeboards in the ICESat-2 sea ice products, The Cryosphere, 15, 821–833, https://doi.org/10.5194/tc-15-821-2021, 2021.

Schneider, T., Lüpkes, C., Dorn, W., Chechin, D., Handorf, D., Khosravi, S., ... & Rinke, A. (2022). Sensitivity to changes in the surface-layer turbulence parameterization for stable conditions in winter: A case study with a regional climate model over the Arctic. Atmospheric Science Letters, 23(1), e1066.

Yu, X., Rinke, A., Dorn, W., Spreen, G., Lüpkes, C., Sumata, H., & Gryanik, V. M. (2020). Evaluation of Arctic sea ice drift and its dependency on near-surface wind and sea ice conditions in the coupled regional climate model HIRHAM–NAOSIM. The Cryosphere, 14(5), 1727-1746.

Martin, T., Tsamados, M., Schroeder, D., & Feltham, D. L. (2016). The impact of variable sea ice roughness on changes in A rctic O cean surface stress: A model study. *Journal of Geophysical Research: Oceans*, *121*(3), 1931-1952

Ricker, R., Fons, S., Jutila, A., Hutter, N., Duncan, K., Farrell, S. L., Kurtz, N. T., and Fredensborg Hansen, R. M.: Linking scales of sea ice surface topography: evaluation of ICESat-2 measurements with coincident helicopter laser scanning during MOSAiC, EGUsphere [preprint], https://doi.org/10.5194/egusphere-2022-1122, 2022.

D. O. Dammann, H. Eicken, A. R. Mahoney, E. Saiet, F. J. Meyer and J. C. "Craig" George, "Traversing Sea Ice—Linking Surface Roughness and Ice Trafficability Through SAR Polarimetry and Interferometry," in IEEE Journal of Selected Topics in Applied Earth Observations and Remote Sensing, vol. 11, no. 2, pp. 416-433, Feb. 2018, doi: 10.1109/JSTARS.2017.2764961.

**Reply to RC2**

**We thank the reviewer for their thorough reading and constructive comments, which has helped to improve the manuscript. In the following, we cite the reviewers' text and add our own answers and text modifications in green.**

**General comments**

The authors present a method to estimate pan-Arctic drag coefficients using observations of sea ice surface feature parameters from ICESat-2. The results show that the drag coefficient is both spatially and temporally variable, and that pan-Arctic drag coefficients can be estimated with the use of satellite observations. It is the first analysis of monthly pan-Arctic drag coefficient estimates of its kind. I assume this will be welcomed in the model community.

Below I will address some specific scientific questions and technical corrections. Based on these, I would like to suggest minor corrections to be made before publication.

My biggest concern is the use of the OIB/ICESat-2 correction and the lack of discussion on the uncertainties and errors this will introduce. The regression is only trained on 4 days of observations in April for a specific location of the Arctic, and is then assumed to still hold over different types of sea ice and other months of the year. I understand there is no more data to use and thus I won't suggest changes to the methods, but I do think a discussion on the downsides of this method is necessary.

> **A discussion exploring these downsides has been added as follows:**
>
> **"How representative the scaling factor is for the whole of the Arctic is difficult to gauge and with limited spatial and temporal near-coincident coverage we expect there to be some uncertainty. Despite these limitations, the racetrack OIB flights from 8 and 12 April 2019 were flown over two distinct ice types. The 8 April racetrack was 100 km north of the Sverdrup Islands (80.5°N) and the 12 April one was centered at 86.5°N in the central Arctic (Kwok et al. 2019a). As a result, the former was over thicker and rougher ice, while the latter was over thinner and smoother ice giving us the opportunity to see how the drag coefficients compare between the two instruments in the different regimes. The scaling factors derived for the two different days are 4.42 and 5.36 respectively, resulting in an uncertainty that is in the range of ±17.5%. This small discrepancy can also be explained by ATL07 sampling: with a smaller obstacle frequency over smooth ice the likelihood of not detecting the few that are present increases (Ricker et al., 2023) thereby increasing the obstacle spacing used in the calculation of drag coefficients for every 10 km segment. Where the obstacle density is generally high, like in rough deformed areas near theCanadian Archipelago, though the detection rate may be low, there will always be an ample amount per 10 km segment to detect a higher**

**drag coefficient signal. Thus, the sampling issue with regard to computing drag coefficients from topography features is more prevalent over smooth ice than rough ice and a higher correction is needed. As 19 and 22 April OIB flights cover larger areas and the rougher deformed ice near the Archipelago is rather small in extent, the scaling factor derived from all 4 days is closer to that of the April 12 racetrack and more representative for the whole Arctic that is predominantly smoother than the ice surveyed on the 8th of April."**

See below for additional comments.

**Specific comments**

- One of the big uncertainties introduced by the methods used in this paper is the OIB model correction to the observed form drag coefficient. This model is trained on the comparison between OIB airborne lidar measurements and ICESat-2 satellite observations for the near-coinciding 4 days in April 2019 in the Lincoln Sea and the Arctic Ocean north of Greenland. This region is for the majority covered in MYI, also in the month these observations were made (see https://nsidc.org/data/nsidc-0611/versions/4).
  This model is then applied to the observations for the full pan-Arctic area discussed in this study, and to each season and for the years 2019, 2020 and 2021. I doubt this relation between the ICESat-2 form drag coefficient and the OIB ATM form drag coefficient will be the same in areas that are predominantly covered in FYI or in other seasons of the year. I understand there are not more near-coincident observations in other regions and months available, so this is the best that can be done now, but I think it is important to include a discussion on the effects these assumptions have on the presented modelled drag coefficient, especially because the model regression coefficient is large and impacts the results a lot.

  **See above.**

- One the same argument, it would be useful to present some statistics on the presented regression model (Eq. 6). How good is the fit? It would also be interesting to see this fit for the observations of the 4 days seperately. Are they similar or does it change for the different days and different flight paths?

  **With regards to the fit, it has been modified to include no y-intercept as the negative y-intercept caused some small form drag coefficients to be negative: which is unrealistic. We now include additional statistics**

(correlation and root means square error between the airborne data and the scaled up satellite data) and discuss them in the text as follows:

"The correlation found between the drag coefficients computed from the different instruments is 0.61 (blue heat map in Fig. 2), and the mean square error (mse) between the OIB ATM drag coefficients and the ICESat-2 ATL07 coefficients with the scaling factor applied (5.28) is 0.11. Considering some ridges are not detected (Ricker et al., 2023) due to sampling issues, and the lack of perfect coincidence, we do not expect perfect correlation. Moreover, we're looking at spatial averages here, where the smoothing has a very strong effect on the ridge spacing (as can be seen in Fig. 1), that is why a topography comparison where the sampling of ICESat-2 is simulated with the OIB ATM data, can show better agreement as in Kwok et al. (2019a). However, that is not our aim in this study, here we try to make the Garbrecht et al. (2002) parameterization applicable to ICESat-2 ATL07 data and correct for the sampling issues using OIB ATM. For comparison's sake, we try to simulate ICESat-2ATL07 with OIB ATM data with the moving average filters in Fig. 2, but we chose not to simulate the elliptical footprint of ICESat-2 in detail as in Kwok et al. (2019a) and Ricker et al. (2023) for that is not needed for the monthly pan-Arctic drag coefficient product which is the end result of this study. Unsurprisingly, comparing the correlation and mse with the OIB ATM data (in blue) to the smoothed version (30 m box [in green] which has the best agreement with the identity line), we have found a correlation of 0.72 and a mse of 0.0024 (with the scaling factor 0.89 as in Fig. 2) for the latter. This better agreement is observed as here the OIB ATM data is sampled similar to how ICESat-2 ATL07 is, making the methods identical will raise the correlation even higher as in Kwok et al. (2019a). What we require for our study is for the drag coefficients to be calculated as in Castellani et al. (2014) and Petty et al. (2017), making use of high resolution and high sampling of the airborne data-sets, and then regressing the OIB ATM values with estimates of the spatially averaged ICESat-2 drag coefficient. In this way, we aim to improve the ICESat-2 product and amplify the signal that is lower than expected due to sampling."

- Explain why the value of 0.2 m is used as threshold (line 153). You've mentioned you have also tested using 0.8 m, but no other values were tried?

  Some cutoff must be introduced to effectively partition skin drag that is associated with centimetre-scale roughness and form drag that is associated to larger obstacles (in this case anything above the 20 cm cutoff), and we chose one which has been used before (e.g., Castellani et al., 2014, Petty et al., 2017) for better a comparison with previous evaluations of Arctic sea ice topography.

- Figure 4A: if you already know this is not a good direct comparison because of the ice drift in between days, maybe it's better to leave this figure out? I think it will only raise doubts and confusion because the fit does not look good, even though you don't really expect it to be good? I think Figure 4B is better because here the drift doesn't influence the comparison.

  **The figure mentioned has been removed.**

- One of the most exciting things of this preprint is the pan-Arctic sea ice roughness dataset it accompanies. I would suggest making this dataset easily accessible: add a link to the data availability statement and include the dataset as an asset on the The Cryosphere page

  **The updated data-set (with y-intercept of the regression set to 0, to avoid negative obstacle form drag values) has been uploaded to PANGAEA Data Publisher for Earth & Environmental Science. The doi leading to the dataset, where individual monthly drag coefficient components are stored for the period 201811-202206, is given both in the references and in the data availability section now.**

Technical corrections

L4. Add 'Ice' to the full name of ICESat-2

   **Corrected**

L5. Replace 'as well airborne surveys' with 'as well as airborne surveys'.

   **Corrected**

L12. I would clarify that it is the drag coefficient of MYI that is above 2.0·10-3

   **Clarified**

L13. I don't understand this last sentence. Do you mean the drag coefficient of this region of MYI is at least 1.5·10-3 everywhere every year?

   **Added clarification**

L22. 'which needs to be better understood': why? There is more discussion of the importance and relevance later in the text, but it would be good to have at least one sentence here to convince the reader this topic is important before going into the more technical details.

**Added following explanation:**

**"By also mapping 3-month aggregates for the years 2019, 2020 and 2021 for better regional analysis, we found the thick multiyear ice area directly north of the Canadian Archipelago and Greenland to be consistently above 2.0 * $10^{-3}$ with the most of the multiyear ice portion of the Arctic typically registering ~ 1.5 * $10^{-3}$ in Spring."**

L32. 'Smoother in comparison' with what?

**Added "rough" in the sentence to make sure it is clearer that "smoother in comparison to multiyear" ice is implied.**

L110. Change 'campaign' to plural: 'campaigns'

**Corrected**

L147. The Rayleigh Criterion introduced here is never explained.

**An appropriate introduction has been added**

L185. Replace ; with 'and'.

**Corrected**

L204. The function to compute the coefficient of resistance might need a reference?

**Reference added**

L271. Change 03 to 0.3

**Corrected**

L347. Change 'a annual' to 'an annual'

**Corrected**

L408. Add figure number

**Corrected**

L436. Change 'first-ice' to 'first-year ice' or 'FYI'

**Corrected**

**References**

Castellani, G. , Lüpkes, C. , Hendricks, S. and Gerdes, R. (2014): Variability of Arctic sea-ice topography and its impact on the atmospheric surface drag , Journal of Geophysical Research: Oceans, 119 (10), pp. 6743-6762 . doi: 10.1002/2013JC009712

Petty, A. A., M. C. Tsamados, and N. T. Kurtz (2017), Atmospheric form drag coefficients over Arctic sea ice using remotely sensed ice topography data, spring 2009–2015, J. Geophys. Res. Earth Surf., 122, doi:10.1002/2017JF004209

---

## Author Response (AR2)

Below we have addressed the final technical corrections of the reviewers (in green).

**Reply to RC1**

-The MSE of 0.11 on line 322, is this in 10^-3? Otherwise it would be very large.

That is correct and this mistake has now been corrected in the text. We thank the reviewer for pointing it out.

**Reply to RC2**

- L314: Awkward wording/sentence structure for the sentence beginning "Obstacle spacing"

Wording simplified to: "With obstacle spacing, the smoothing gets in the way of extracting any meaningful relationship."

- Figure 4: I don't know if both panels are necessary.

Both panels were kept in the end, since we think having both together offers the most information on the small changes visible in the normalized version and the difference in size in the non-normalized version.

- L464: Section title for 3.6 doesn't convey the contents of that section very well to me.

Section title changes to "Discussion and concluding remarks".

- L504: missing/broken reference link

The link has been fixed. We thank the reviewer for pointing this out.

- L649: Check figure reference "Fig. A3C and Fig. A3D". Should this be B and C instead of C and D?

The text has been corrected. We thank the reviewer for pointing this out.

- Figure A4: missing/broken reference link in figure caption

The link has been fixed. We thank the reviewer for pointing this out.

We thank both reviewers for their time and overall improvement of the manuscript.